# ANALYTIC CONTINUAL TEST-TIME ADAPTATION FOR MULTI-MODALITY CORRUPTION

## ABSTRACT

Test-Time Adaptation (TTA) aims to help pre-trained model bridge the gap between source and target datasets using only the pre-trained model and unlabelled test data. A key objective of TTA is to address domain shifts in test data caused by corruption, such as weather changes, noise, or sensor malfunctions. Multi-Modal Continual Test-Time Adaptation (MM-CTTA), an extension of TTA with better real-world applications, further allows pre-trained models to handle multi-modal inputs and adapt to continuously-changing target domains. MM-CTTA typically faces challenges including **error accumulation**, **catastrophic forgetting**, and **reliability bias**, with few existing approaches effectively addressing these issues in multi-modal corruption scenarios. In this paper, we propose a novel approach, Multi-modality Dynamic Analytic Adapter (MDAA), for MM-CTTA tasks. We innovatively introduce analytic learning into TTA, using the Analytic Classifiers (ACs) to prevent model forgetting. Additionally, we develop Dynamic Selection Mechanism (DSM) and Soft Pseudo-label Strategy (SPS), which enable MDAA to dynamically filter reliable samples and integrate information from different modalities. Extensive experiments demonstrate that MDAA achieves state-of-the-art performance on MM-CTTA tasks while ensuring reliable model adaptation.

## 1 INTRODUCTION

Test-Time Adaptation (TTA) aims to help the pre-trained model bridge the gap between the source domain and the target domain (Wang et al., 2021; Liang et al., 2024). Unlike Unsupervised Domain Adaptation (UDA) (Zhang et al., 2015; Liang et al., 2024), TTA performs adaptation without the need for any source data (*i.e.,* pre-trained dataset), which not only saves computational resources by avoiding retraining but also preserves the privacy of the source data. One key TTA application is addressing the problem of domain shift from source data to corrupted test data, where the corruption is often caused by external factors (*e.g.,* weather changes, ambient noise) or sensor malfunctions. As an extension of TTA, Continual Test-Time Adaptation (CTTA) has been proposed to align with real-world scenarios where domain shifts usually are dynamic (Wang et al., 2022). Challenges in CTTA mainly consist of **error accumulation** and **catastrophic forgetting**. **Error accumulation**, stemming from incorrect pseudo-labels, can mislead models' adaptation and potentially lead to collapse (Chen et al., 2019). **Catastrophic forgetting** refers to the loss of knowledge from the source data during continuous adaptation, reducing the model's generalization ability (McCloskey & Cohen, 1989). To address these challenges, various CTTA methods have been proposed, yielding promising results in corruption-related tasks (Wang et al., 2022; Gao et al., 2023; Niu et al., 2022).

Most existing CTTA approaches focus solely on single-modal scenarios, paying less attention to multi-modal applications. Compared with TTA and CTTA, Multi-Modal Continual Test-Time Adaptation (MM-CTTA) (Cao et al., 2023) shows greater potential for real-world applications, as multi-modal data integrates a broader range of information than single-modality adaptation, resulting in more robust networks (Radford et al., 2021). However, applying existing CTTA methods to MM-TTA by simply replacing the backbone with muiti-modal encoders is less optimal. MM-CTTA performance can easily suffer from **reliability bias**, where intra-modality domain shifts increase information discrepancies in downstream fusion networks (Yang et al., 2024). Such effect becomes more pronounced when modality corruption changes dynamically. Although a few works address Multi-Modal Test-Time Adaptation (MM-TTA) (Yang et al., 2024; Lei & Pernkopf, 2024), they

struggle with model forgetting during continuous adaptation and the challenges posed by interleaved modality corruption.

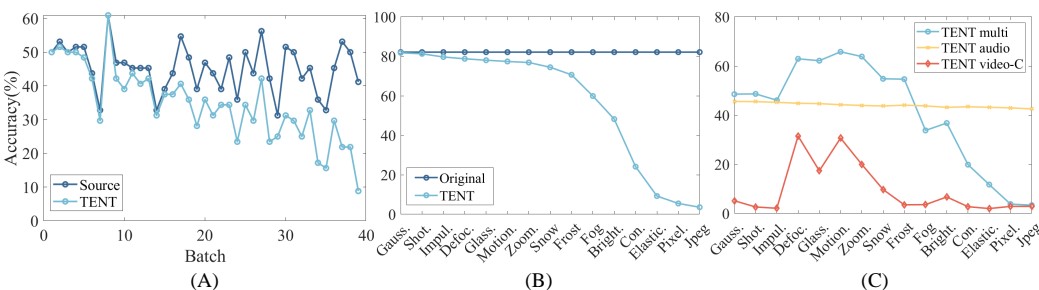

Figure 1: Illustration of the three key challenges in MM-CTTA, using the representative TTA method TENT (Wang et al., 2021) with CAV-MAE (Gong et al., 2022b). **(A) Error accumulation**: The model's performance progressively degrades as adaptation progresses (batch increases). This experiment is conducted on the Kinetic-C Fog dataset (Kay et al., 2017; Yang et al., 2024), with the "Source" representing the model without adaptation during inference. **(B) Catastrophic forgetting**: The model's performance on the source data significantly declines during continuous adaptation. The "Original" refers to the model's performance on the clean test set of Kinetics50 without any adaptation, while "TENT" represents the performance on the same dataset after adaptation to the corresponding corruption. **(C) Reliability bias**: As the dominant modality (video) becomes increasingly corrupted, the performance of the multi-modal network deteriorates, even falling below that of the audio-only network. This experiment is conducted on Kinetic-C, with video as the dominant corrupted modality.

To further illustrate, we use a representative TTA method, TENT (Wang et al., 2021), as an example in Fig.1. The results indicate that traditional TTA methods face significant limitations due to **error accumulation**, **catastrophic forgetting**, and **reliability bias**. These challenges suggest that methods designed for MM-CTTA must meet three key requirements: (1) effectively mitigating error accumulation during adaptation, (2) retaining knowledge of the source data after adapting to various domains, and (3) dynamically suppressing the influence of corrupted modalities while prioritizing more reliable ones. To the best of our knowledge, the method closest to meeting these criteria is CoMAC (Cao et al., 2023), which focuses on segmentation tasks in living environments rather than addressing modality corruption.

In this paper, we propose a new approach named the **M**ulti-modality **D**ynamic **A**nalytic **A**daptor (MDAA) to address the challenges in MM-CTTA. MDAA comprises three primary components: (i) the Analytic Classifiers (ACs), (ii) the Dynamic Selection Mechanism (DSM), and (iii) the Soft Pseudo-label Strategy (SPS). ACs update the model by addressing a recursive ridge regression problem, optimizing on both new target data and learned knowledge to avoid **catastrophic forgetting**. DSM selectively updates each AC based on its output reliability, thereby alleviating **reliability bias**. The SPS enhances the model's robustness to label noise by assigning varying probabilities to multiple labels, which mitigates potential **error accumulation**. The key contributions of this work can be summarised as follows:

1). We propose a method, MDAA, for a more challenging TTA task named MM-CTTA, and explain why typical TTA methods are not well-suited for this task, as illustrated by a example in Fig.1.

2). We innovatively apply AC to TTA to keep the model from **catastrophic forgetting** during adaption. We propose DSM and SPS to further dynamically integrate the features of different modal thus mitigating **reliability bias** the **error accumulation**.

3). We design two MM-CTTA tasks to meet the needs of real-world environments. Extensive experiments demonstrate that MDAA achieves SOTA performance, surpassing previous methods by up to 6.22% and 6.84% in the two tasks respectively.

## 2 RELATED WORKS

### 2.1 TEST-TIME ADAPTATION

**Test-Time Adaptation** (TTA) focuses on enabling a pre-trained model to adapt to a new target domain without requiring access to the source domain data used to initially train the model. A major challenge in TTA is error accumulation, which affects methods that rely on pseudo-labeling; incorrect predictions can mislead the adaptation process, potentially causing the model to collapse (Chen et al., 2019). One of the early solutions, TENT, addresses this by only updating the model's batch normalization (BN) layers through entropy minimization (Wang et al., 2021). Subsequent research (Niu et al., 2023; Gong et al., 2022a; Zhou et al., 2023) has further mitigated this issue by filtering out low-confidence predictions using carefully designed thresholds and updating the layer normalization (LN) layers for more robust performance.

### 2.2 CONTINUAL TEST-TIME ADAPTATION

**Continual Test-Time Adaptation** (CTTA) extends TTA to scenarios where the target domain changes continuously in an online manner without access to source data. The additional challenge CTTA faces is known as catastrophic forgetting, which occurs when a model adapts to the target domain, leading to the loss of knowledge acquired from the source domain and dimension of generalization ability. To solve this problem, some studies turn to Continual Learning (CL) and achieve great success (Niu et al., 2022; Cao et al., 2023). We follow this trend and introduce a novel CL approach called Analytic Continual Learning (ACL) in MDAA.

ACL provides a global optimal solution through matrix inverse operations (Guo & Lyu, 2004). To address the out-of-memory issue caused by large inverse matrices, Zhuang et al. (2021a) demonstrated that iterative computation using block-wise data achieves results equivalent to joint computation, making analytic learning highly effective in continual learning. By treating data from different time periods as blocks, ACL allows for recursive computation, with the final result being as accurate as if all data were processed simultaneously. Thanks to its non-forgetting properties, ACL has shown strong performance across various CL tasks in recent years (Zhuang et al., 2022; 2023). Inspired by ACL's success in CL tasks, we apply ACL to TTA for the first time in this work, implementing several enhancements to address the issue of catastrophic forgetting.

### 2.3 MULTI-MODALITY TEST-TIME ADAPTATION

**Multi-modality Test-Time Adaptation** (MM-TTA) seeks to enhance model reliability by incorporating multi-modal data into the TTA task. However, imbalances in inter-modal reliability can result in significant performance degradation, known as reliability bias (Wang et al., 2020). Most existing MM-TTA models address this issue by independently updating BN or LN layers of each feature encoder, followed by a weighted fusion mechanisms (Shin et al., 2022; Cao et al., 2023; Xiong et al., 2024). Although this allows more reliable modalities to carry more weight during fusion, the approach remains relatively shallow in terms of information integration.

A recent model called READ (Yang et al., 2024) introduced a more advanced approach by fusing features through a Vision Transformer (ViT) (Dosovitskiy et al., 2020) block, which allows for the preservation of parameters inherited from source data while effectively integrating inter-modal information (Vaswani, 2017; Gong et al., 2022b). READ achieves reliable adaptation by modulating only the fusion layer within the attention module of the ViT block. Follow-up work (Lei & Pernkopf, 2024) aimed to improve performance by updating both the feature encoders and the fusion layer. However, this approach requires prior knowledge about which modalities are corrupted, making it less capable for real-world scenario.

In this paper, our MDAA approach introduces classifiers for each feature encoder and the fusion network. By adjusting the parameters of each classifier individually, MDAA maximizes the use of information from each modality, thus mitigating the effects of reliability bias. Crucially, these classifier updates for upstream and downstream blocks do not cause conflicts, as the entire pre-trained model remains frozen throughout the process. This property enables MDAA to adapt to changing modality corruption without extra prior knowledge, distinguishing it from existing MM-TTA approaches.

## 3 METHOD

In this section, we first define the challenging MM-CTTA setting and introduce notations for key concepts in Sec. 3.1. We then introduce the proposed Multi-modality Dynamic Analytic Adapter (MDAA) approach, which includes Analytic Classifiers (ACs) integrated with pre-trained multi-modal encoders (Sec. 3.2), the Dynamic Selection Mechanism (DSM) (Sec. 3.3), and the Soft Pseudo-label Strategy (SPS) (Sec. 3.4). An overview of MDAA is illustrated in Fig. 2, and the pseudo-code for MDAA is provided in Appendix B.

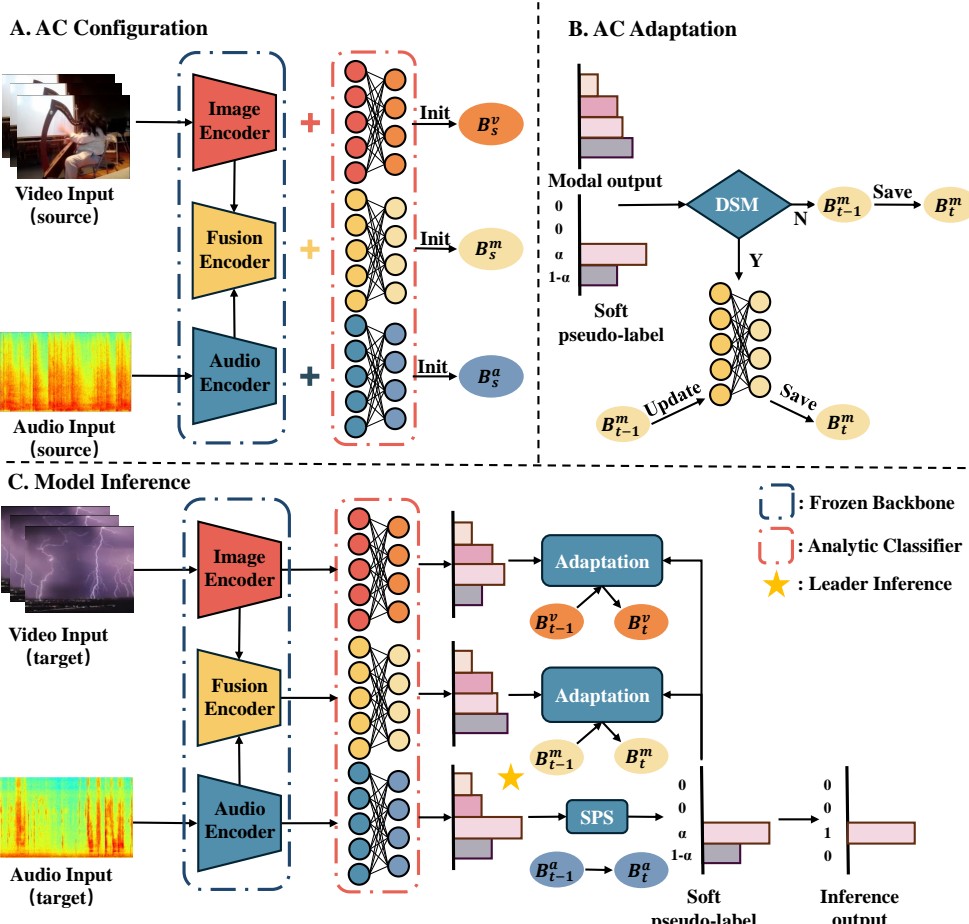

Figure 2: The overview of MDAA. (A) Construct Analytic Classifiers (ACs) and initialize the memory bank to preserve the knowledge of source datasets. (B) Use the Dynamic Selection Mechanism (DSM) to determine if the classifier and memory bank require updates. (C) Generate pseudo-labels for adaptation and inference. The Soft Pseudo-label Strategy (SPS) is shown using top 2 largest probability labels.

### 3.1 PROBLEM DEFINITION AND NOTATIONS

In this paper, we focus on an audio-video classification task as an example, using two modalities for illustration without loss of generality. In MM-CTTA, the pre-trained model $\Phi_S(\cdot)$ is trained on a labeled source dataset $D_S \sim \{\mathbf{X}_S^a, \mathbf{X}_S^v, \mathbf{Y}_S\}$ in source domain S, where $\mathbf{X}_S^a$ and $\mathbf{X}_S^v$ represent the audio and video training data matrices respectively. $\mathbf{Y}_S$ represents the corresponding one-hot label set. During the adaptation phase, for each timestamp $t$ in the target domain T, the model must perform inference and update its parameters based on the unlabeled test dataset $D_{T,t} \sim \{\mathbf{X}_{T,t}^a, \mathbf{X}_{T,t}^v\}$. The suffix $T, t$ indicates the current target domain, as it shifts continually. Note that only the test dataset at timestamp $t$ is available for updating the parameters $\Phi_{T,t} \to \Phi_{T,t+1}$.

## 3.2 SOURCE MODEL AND ANALYTIC CLASSIFIER CONFIGURATION

In the context of multi-modality classification task, we propose to integrate the ACs as classifiers into a typical extraction-fusion approach. The standard structure can be represented as follows:

$$\Phi_S\left(\mathbf{X}^a, \mathbf{X}^v\right) = \eta_S\left(f_S^m\left(f_S^a\left(\mathbf{X}^a\right) \otimes f_S^v\left(\mathbf{X}^v\right)\right)\right), \tag{1}$$

where $f_S^a\left(\cdot\right)$ and $f_S^v\left(\cdot\right)$ represent upstream feature encoders for audio and video modalities, respectively, $\otimes$ indicates the fusion operation implemented as concatenation in this paper, $f_S^m\left(\cdot\right)$ denotes the downstream fusion network and $\eta_S\left(\cdot\right)$ indicates the classifier.

Specifically, we leverage multiple ACs as classifiers for extracted features of each modality and the fused features, with each classifier making an independent prediction. Each AC is of the same structure as a two-layer fully connected network, denoted by $\zeta\left(FE\left(\cdot\right)\right)$. Features are first non-linearly projected into a higher dimensional space $\varphi$ by the feature expansion layer $FE\left(\cdot\right)$, enhancing their expressiveness (Zhuang et al., 2022; 2021b). The projected features are then passed through the linear layer $\zeta\left(\cdot\right)$ for classification. The feature expansion layer remains frozen, while the linear layer requires updates during the adaptation process. For simplicity, we combine the feature encoder $f_S\left(\cdot\right)$ and the feature expansion layer $FE\left(\cdot\right)$ into a single function, denoted as $f_S'\left(\cdot\right)$, and refer to the projected features as $X_{\text{exf}}$. Consequently, MDAA yields three classifiers as follows:

$$\Phi_S^i\left(\mathbf{X}^i\right) = \zeta_S^i\left(f_S^{i'}\left(\mathbf{X}^i\right)\right) = \zeta_S^i\left(\mathbf{X}_{\text{exf}}^i\right), i \in \{a, v\}, \tag{2}$$

$$\Phi_S^m\left(\mathbf{X}^a, \mathbf{X}^v\right) = \zeta_S^m\left(f_S^{m'}\left(f_S^a\left(\mathbf{X}^a\right) \otimes f_S^v\left(\mathbf{X}^v\right)\right)\right) = \zeta_S^m\left(\mathbf{X}_{\text{exf}}^m\right). \tag{3}$$

Since all ACs can be represented by the same equation, we will not distinguish between them except for special needs in the following discussion. Following previous ACL works Zhuang et al. (2022), the classifier $\zeta_S$ are updated by solving the ridge regression to optimize on the source data $D_S$. To further solve the class imbalance problem within the source data, inspired by Fang et al. (2024), we formulate the optimization problem as follows:

$$\underset{\mathbf{W}_S}{\arg\min} \quad \sum_{k=1}^{N_S} \omega_k \left\|\mathbf{y}_{S,k} - \mathbf{x}_{\text{exf},k}\mathbf{W}_S\right\|_F^2 + \gamma \left\|\mathbf{W}_S\right\|_F^2, \tag{4}$$

where $\left\|\cdot\right\|_F$ indicates the Frobenius norm, $\gamma$ is the regularization parameter and $N_S$ is the samples size of $D_S$. $\omega_k$, $\mathbf{x}_{\text{exf},k}$ and $\mathbf{y}_{S,k}$ represents the weight, expanded feature vector and one-hot label of sample k in $D_S$, while the weight is further defined as

$$\omega_k = \frac{N_S}{N_C \times N_{c|k}}, \tag{5}$$

where $N_C$ is the number of classes in $D_S$ and $N_{c|k}$ is the number of samples in $D_S$ from category $c$ to which sample $k$ belongs. Following the ridge regression solution, the solution to optimization problem is given in **Theorem 1**.

**Theorem 1.** The optimal solution to Formula 4 is given as

$$\hat{\mathbf{W}}_S = \left(\sum_{k=1}^{N_S} \tilde{\mathbf{x}}_{\text{exf},k}^\top \tilde{\mathbf{x}}_{\text{exf},k} + \gamma\mathbf{I}\right)^{-1} \sum_{k=1}^{N_S} \tilde{\mathbf{x}}_{\text{exf},k}^\top \tilde{\mathbf{y}}_{S,k}$$

$$= \left(\tilde{\mathbf{X}}_{\text{exf},S}^\top \tilde{\mathbf{X}}_{\text{exf},S} + \gamma\mathbf{I}\right)^{-1} \tilde{\mathbf{X}}_{\text{exf},S}^\top \tilde{\mathbf{Y}}_S, \tag{6}$$

where $\tilde{\mathbf{x}}_{\text{exf},k} = \sqrt{\omega_k}\mathbf{x}_{\text{exf},k}$ and $\tilde{\mathbf{y}}_{S,k} = \sqrt{\omega_k}\mathbf{y}_{S,k}$. The proof of Theorem 1 is provided in Appendix A. In addition to the classifier weights $W_S$, a memory bank $B_S$ needs to be constructed during the training phase. Unlike other methods (Cao et al., 2023; Zhang et al., 2023; Xiong et al., 2024), our memory bank contains only two types of matrices, which can be represented as $B_S \sim \{\mathbf{P}_S, \mathbf{P}_S\}$, where

$$\mathbf{Q}_S = \tilde{\mathbf{X}}_{\text{exf},S}^\top \tilde{\mathbf{X}}_{\text{exf},S} + \gamma\mathbf{I}, \tag{7}$$

$$\mathbf{Q}_S = \tilde{\mathbf{X}}_{\text{exf},S}^\top \tilde{\mathbf{Y}}_S. \tag{8}$$

Both $\mathbf{P}_S$ and $\mathbf{Q}_S$ are used to extract and preserve the learned knowledge from the source dataset, which cannot be accessed during adaptation. Therefore weight $\hat{\mathbf{W}}_S$ can be further rewritten as

$$\hat{\mathbf{W}}_S = \mathbf{P}_S^{-1}\mathbf{Q}_S. \tag{9}$$

### 3.3 DYNAMIC SELECTION MECHANISM

When adapting to sequentially incoming multi-modal data in MM-CTTA, some modality may be unreliable due to corruption. Such corrupted data can mislead the model to learn incorrect information. To address the reliability bias, we propose Dynamic Selection Mechanism (DSM) to determine whether each AC should be updated in a dynamic way. DSM first identifies the most reliable classifier among the three (*i.e., a, v, m*) as the *leader*, while the other classifiers are treated as *follower*.

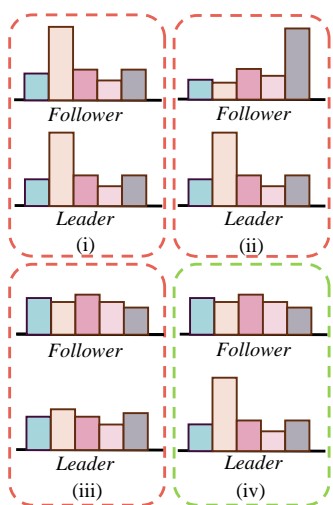

Specifically, the *leader* is determined by comparing the maximum probability from each classifier's distribution. For each sample, the model prediction corresponds to the *leader*'s prediction, and the *leader* will not be updated to maintain class balance. Whether to update each *follower* in the model depends on the comparison between its maximum probability distribution $\max P(Follower)$ and the *leader*'s maximum probability distribution $\max P(Leader)$. We consider four possible scenarios, as illustrated in Fig.3:

(i). **Close Distributions**: $\max P(Leader)$ and $\max P(Follower)$ are quite close and refer to the same label. In this case, the *follower* is not updated, as this would only reinforce what it has already learned, potentially leading to an imbalanced class distribution.

(ii). **Different Labels with Close Probabilities**: $\max P(Leader)$ and $\max P(Follower)$ are quite close while they refer to different labels. The *follower* is also not updated in this scenario because there is no certainty that the *leader*'s result is correct, and updating may introduce errors.

(iii). **Evenly Distributed Probabilities**: Both the *leader* and *follower* have evenly distributed probabilities with no significant difference between labels. Again, no update occurs.

(iv). **Significant Difference**: The *leader* has a higher probability label, while the *follower* has a more even probability distribution. In this scenario, the *follower* should be updated.

Figure 3: Four possible relationship between the probability distribution of *leader* and *follower*. Only samples belong to case (iv) will be used to update *follower*.

In general, cases (i),(ii) and (iii) all belong to a small gap between $\max P(Follower)$ and $\max P(Leader)$, while case (iv) belongs to a larger gap. Therefore, given a pre-defined threshold $\theta$, DSM can be noted as:

$$\begin{cases} \text{Accept}, & \max P(Leader) - \max P(Follower) \geqslant \theta \\ \text{Reject}, & \text{otherwise} \end{cases} \tag{10}$$

### 3.4 SOFT PSEUDO-LABEL STRATEGY

Using soft labels in self-supervised learning is quite popular as it preserves a broader range of possible outcomes compared to hard (one-hot) label learning (Müller et al., 2019; Hinton et al., 2015). Inspired by such trend, we use soft pseudo-labels to update the ACs during MM-CTTA, as illustrated in Fig.2(C). For each test sample $k$, we choose the top $n$ classes of *leader*'s distribution set $C = \{c_{1,k}, c_{2,k}, \ldots, c_{n,k}\}$ and assign them with weights $\alpha_{1,k}, \alpha_{2,k} \ldots \alpha_{n,k}, (\sum_{i=1}^{n} \alpha_{i,k} = 1)$ respectively. The reconstructed label $\bar{y}$ for sample $k$ through SPS can be represented as

$$\bar{\mathbf{y}}_k = \begin{cases} \alpha_{i,k} & , i \in C \\ 0 & , \text{otherwise} \end{cases} \tag{11}$$

Since ACL considers global optimization, which means it accounts not only for the input data $\mathbf{X}_{\mathrm{T}}, t$ at timestamp $t$, but also all previous data processed by the model, including $\mathbf{X}_{\mathrm{S},k}$ and $\mathbf{X}_{\mathrm{T},1:t-1}$. With the reconstructed label $\mathbf{Y}_{\mathrm{T},1:t}$ throughout $t$, the optimization problem for weight matrix $\mathbf{W}_{\mathrm{T},t}$ can be represented as:

$$\underset{\mathbf{W}_{\mathrm{T},t}}{\arg\min} \quad \sum_{k=1}^{N_{\mathrm{S}}} \omega_k \left\| \mathbf{y}_{\mathrm{S},k} - \mathbf{x}_{\mathrm{exf},k} \mathbf{W}_{\mathrm{T},t} \right\|_{\mathrm{F}}^2 + \left\| \bar{\mathbf{Y}}_{\mathrm{T},1:t} - \mathbf{X}_{\mathrm{exf},1:t} \mathbf{W}_{\mathrm{T},t} \right\|_{\mathrm{F}}^2 + \gamma \left\| \mathbf{W}_{\mathrm{T},t} \right\|_{\mathrm{F}}^2. \tag{12}$$

It is important to note that while we treat source data and target data separately, we adjust the category balance for the source dataset by assigning weights to each class. In contrast, it is challenging

to apply this approach to the target dataset, as it is impossible to know the exact number of samples in each category within the target domain. However, with the help of the DSM, category balance can still be maintained, as unimportant samples have been filtered out. Given that the weights added to the sample data are expected to average out to 1, the impact of each sample from both the source and target domains on the model can be considered equivalent.

Due to the definition of TTA, all datasets (*i.e.,* $\mathbf{X}_{\text{S},k}$ and $\mathbf{X}_{\text{T},1:t-1}$) prior to timestamp $t$ are not accessible when solving the optimization problem. However, with the aid of the memory bank $B_{\text{T},t} \sim \{\mathbf{P}_{\text{T},t}, \mathbf{Q}_{\text{T},t}\}$, the solution can still be computed, as stated in **Theorem 2.**

**Theorem 2.** The optimal solution to Formula 12 is given as

$$\hat{\mathbf{W}}_{\text{T},t} = (\tilde{\mathbf{X}}_{\text{exf,S}}^{\top}\tilde{\mathbf{X}}_{\text{exf,S}} + \mathbf{X}_{\text{exf},1:t}^{\top}\mathbf{X}_{\text{exf},1:t} + \gamma\mathbf{I})^{-1}(\tilde{\mathbf{X}}_{\text{exf,S}}^{\top}\tilde{\mathbf{Y}}_{\text{S}} + \mathbf{X}_{\text{exf},1:t}^{\top}\bar{\mathbf{Y}}_{\text{T},1:t}) = \mathbf{P}_{\text{T},t}^{-1}\mathbf{Q}_{\text{T},t}, \quad (13)$$

where the memory bank is updated in a recursive way with timestamp $t$, as

$$\mathbf{P}_{\text{T},t} = \mathbf{P}_{\text{S}} + \mathbf{X}_{\text{exf},1:t}^{\top}\mathbf{X}_{\text{exf},1:t} = \mathbf{P}_{\text{T},1} + \mathbf{X}_{\text{exf},2:t}^{\top}\mathbf{X}_{\text{exf},2:t}$$

$$= \cdots = \mathbf{P}_{\text{T},t-1} + \mathbf{X}_{\text{exf},t}^{\top}\mathbf{X}_{\text{exf},t} \quad (14)$$

$$\mathbf{Q}_{\text{T},t} = \mathbf{Q}_{\text{S}} + \mathbf{X}_{\text{exf},1:t}^{\top}\bar{\mathbf{Y}}_{\text{T},1:t} = \mathbf{Q}_{\text{T},1} + \mathbf{X}_{\text{exf},2:t}^{\top}\bar{\mathbf{Y}}_{\text{T},2:t}$$

$$= \cdots = \mathbf{Q}_{\text{T},t-1} + \mathbf{X}_{\text{exf},t}^{\top}\bar{\mathbf{Y}}_{\text{T},t}. \quad (15)$$

The *proof* of **Theorem 2** is also provided in Appendix A. It can be seen that, the size of the memory bank $B$ is constant and depends on the dimension of the feature expansion layer $\varphi$ and class number $N_C$. The equations above demonstrate that the size of $B$ remains unchanged during adaptation, regardless of the timestamp $t$.

## 4 EXPERIMENTS

In this section, we evaluate the proposed MDAA approach under the challenging MM-CTTA setting. The MM-CTTA setting is detailed in Sec. 4.1. Sec. 4.2 compares MDAA with SOTA methods through extensive experiments to demonstrate its superior performance. Additionally, the ablation studies on each component of MDAA are presented in Sec. 4.3, which illustrate the effectiveness of MDAA in addressing different challenges within the MM-CTTA setting. The implementation details are provided in Appendix C.

### 4.1 BENCHMARKS AND SETTINGS

In this section, we introduce the datasets and task settings used for MM-CTTA. The MM-CTTA setting requires the model to initially train on uncorrupted source datasets. Subsequently, the model performs TTA on each corrupted target domain in sequence. We utilize two datasets for this setting: Kinetics50 (Kay et al., 2017) and VGGSound (Chen et al., 2020). While the original uncorrupted datasets serve as source, the corrupted target datasets, Kinetics50-C and VGGSound-C, are constructed following Yang et al. (2024), which introduces 15 types of video corruptions and 6 audio corruptions at severity level 5.

To evaluate the model performance, we designed two classification tasks specifically for MM-CTTA following previous research on corruption-related TTA (Wang et al., 2022; 2021; Yang et al., 2024; Lei & Pernkopf, 2024). The first task, named **progressive single-modality corruption**, sequentially introduces different types of corruption to one modality while keeping the other modality uncorrupted. Focusing on evaluating the model's resistance to catastrophic forgetting, this task is set in an online manner, where the model processes only one sample at a time. The second task, called **interleaved modality corruption**, continually alternates corruption between the two modalities. While most methods perform poorly in the online setting due to severe catastrophic forgetting, this task uses a batch size of 64 during test time to emphasize assessing the model's ability to adapt to dynamic reliability biases.

### 4.2 PERFORMANCE COMPARISON

To provide a comprehensive comparison, we reproduce different types of TTA methods under the MM-CTTA setting. Typical TTA methods of TENT (Wang et al., 2021) and SAR (Niu et al., 2023);

Table 1: Comparison with SOTA methods on **audio progressive single-modality corruption** task in terms of classification Top-1 accuracy (%), using dataset **Kinetics50-C** and **VGGSound-C** in severity level 5. The best results for each domain are highlighted in **bold**. * means we revise the method from BN to LN for fair comparison.

| Method | Type | Kinetics50-C | | | | | | | VGGSound-C | | | | | | |
|---|---|---|---|---|---|---|---|---|---|---|---|---|---|---|---|
| | | Gauss. | Traff. | Crowd | Rain | Thund. | Wind | Avg. | Gauss. | Traff. | Crowd | Rain | Thund. | Wind | Avg. |
| | | $t$ | | | | | $\longrightarrow$ | | $t$ | | | | | $\longrightarrow$ | |
| Source | - | **73.97** | 65.17 | 67.88 | 70.24 | 68.00 | 70.44 | 69.28 | 37.29 | 21.24 | 16.89 | 21.81 | 27.36 | 25.66 | 25.04 |
| TENT* | TTA | 73.02 | 63.36 | 45.31 | 37.02 | 34.57 | 34.01 | 47.88 | 0.68 | 0.28 | 0.28 | 0.28 | 0.28 | 0.28 | 0.35 |
| SAR | TTA | 72.18 | 70.36 | 48.30 | 37.67 | 36.21 | 39.09 | 50.64 | 16.09 | 4.50 | 4.33 | 3.60 | 12.00 | 5.51 | 7.67 |
| CoTTA | CTTA | 19.67 | 4.10 | 2.11 | 2.03 | 2.03 | 2.03 | 5.33 | 5.85 | 1.35 | 0.52 | 0.53 | 0.57 | 0.38 | 1.53 |
| EATA | CTTA | 73.91 | 65.29 | 68.24 | 70.51 | 68.28 | 70.48 | 69.45 | **40.39** | 31.99 | 31.91 | 32.38 | **39.24** | 33.95 | 34.98 |
| MMTTA* | MM-TTA | 17.03 | 1.99 | 1.99 | 1.99 | 1.99 | 1.99 | 4.50 | 0.41 | 0.33 | 0.33 | 0.33 | 0.33 | 0.33 | 0.34 |
| READ | MM-TTA | 68.33 | 59.75 | 57.38 | 54.14 | 53.49 | 52.72 | 57.63 | 18.53 | 7.99 | 7.44 | 5.71 | 8.19 | 4.73 | 8.77 |
| MDAA | MM-CTTA | 72.87 | **71.45** | **72.91** | **72.26** | **73.20** | **73.80** | **72.75** | 38.80 | **34.91** | **34.63** | **34.59** | 37.70 | **35.85** | **36.08** |
| | | $t$ | | | | | $\longleftarrow$ | | $t$ | | | | | $\longleftarrow$ | |
| TENT* | TTA | 43.27 | 42.96 | 43.81 | 60.19 | 69.17 | 70.17 | 54.93 | 0.30 | 0.30 | 0.30 | 0.30 | 0.30 | 0.39 | 0.32 |
| SAR | TTA | 41.81 | 27.94 | 24.41 | 40.47 | 42.90 | 70.36 | 41.32 | 14.91 | 4.56 | 4.61 | 3.72 | 12.44 | 5.94 | 7.70 |
| CoTTA | CTTA | 1.99 | 1.99 | 2.92 | 5.07 | 11.64 | 32.56 | 9.36 | 0.30 | 0.30 | 0.30 | 0.30 | 0.30 | 0.36 | 0.31 |
| EATA | CTTA | 73.91 | 65.32 | 68.18 | 70.49 | 68.26 | 70.47 | 69.44 | **40.22** | 33.69 | 31.61 | 32.64 | **39.67** | 32.81 | 35.11 |
| MMTTA* | MM-TTA | 1.99 | 1.99 | 1.99 | 1.99 | 1.99 | 39.21 | 8.19 | 0.33 | 0.30 | 0.31 | 0.30 | 0.61 | 1.49 | 0.56 |
| READ | MM-TTA | 56.50 | 56.09 | 56.30 | 57.25 | 62.99 | 65.14 | 59.05 | 9.20 | 5.82 | 7.48 | 7.89 | 11.67 | 12.11 | 9.03 |
| MDAA | MM-CTTA | **74.86** | **72.63** | **72.87** | **72.26** | **73.4** | **72.02** | **73.01** | 38.95 | **35.46** | **34.66** | **34.70** | 37.31 | **35.20** | **36.05** |

Table 2: Comparison with SOTA methods on **video progressive single-modality corruption** task in terms of classification Top-1 accuracy (%), with dataset **Kinetics50-C** in severity level 5.

| Method | Type | Gauss. | Shot. | Impul. | Defoc. | Glass. | Motion. | Zoom. | Snow | Frost | Fog | Bright. | Cont. | Elastic. | Pixel. | Jpeg | Avg. |
|---|---|---|---|---|---|---|---|---|---|---|---|---|---|---|---|---|---|
| | | | | | $t$ | | | | | | $\longrightarrow$ | | | | | | |
| Source | - | 48.74 | 49.80 | 48.99 | 67.68 | 61.84 | 70.88 | 66.18 | 61.35 | 61.39 | 45.34 | 75.95 | 51.87 | 65.77 | 68.78 | 66.10 | 60.71 |
| TENT* | TTA | 16.23 | 2.07 | 2.03 | 2.08 | 2.06 | 2.03 | 2.03 | 2.03 | 2.03 | 2.03 | 2.03 | 2.03 | 2.03 | 2.03 | 2.03 | 2.98 |
| SAR | TTA | 38.36 | 35.97 | 34.51 | 44.40 | 48.86 | 50.77 | 47.53 | 43.59 | 35.81 | 42.54 | 52.11 | 35.44 | 50.20 | 40.15 | 50.73 | 43.40 |
| CoTTA | CTTA | 33.43 | 27.51 | 25.20 | 21.19 | 18.19 | 16.41 | 14.91 | 13.29 | 11.18 | 9.60 | 8.43 | 6.89 | 6.36 | 5.39 | 4.09 | 14.80 |
| EATA | CTTA | 48.80 | 49.82 | 49.03 | 67.66 | 61.98 | 70.84 | 66.16 | 61.64 | 61.54 | 45.40 | **75.99** | 51.95 | 65.88 | 68.71 | 66.08 | 60.77 |
| MMTTA* | MM-TTA | 14.31 | 2.64 | 2.03 | 2.03 | 2.03 | 2.03 | 2.03 | 2.03 | 2.03 | 2.03 | 2.03 | 2.03 | 2.03 | 2.03 | 2.03 | 2.89 |
| READ | MM-TTA | 11.92 | 2.04 | 2.03 | 2.97 | 2.41 | 2.46 | 2.41 | 2.30 | 2.04 | 2.04 | 2.04 | 2.04 | 2.04 | 2.04 | 2.03 | 2.86 |
| MDAA | MM-CTTA | 54.89 | 55.25 | 55.32 | 63.89 | 62.49 | 67.26 | 65.86 | 64.32 | 65.31 | 61.86 | 73.20 | 61.60 | 67.83 | 69.22 | 68.69 | 63.80 |
| | | | | | $t$ | | | | | | $\longleftarrow$ | | | | | | |
| TENT* | TTA | 2.03 | 2.03 | 2.03 | 2.03 | 2.03 | 2.03 | 2.03 | 2.03 | 2.03 | 2.03 | 2.03 | 2.03 | 2.10 | 3.47 | 53.26 | 5.55 |
| SAR | TTA | 34.75 | 35.08 | 35.89 | 42.70 | 45.99 | 49.43 | 50.12 | 44.08 | 42.42 | 40.02 | 57.54 | 35.56 | 48.86 | 57.22 | 66.26 | 45.73 |
| CoTTA | CTTA | 3.89 | 4.01 | 4.50 | 5.39 | 5.47 | 5.75 | 7.82 | 4.98 | 5.79 | 9.85 | 6.89 | 12.72 | 14.26 | 22.33 | 51.34 | 11.00 |
| EATA | CTTA | 48.81 | 49.79 | 49.02 | 67.71 | 61.96 | **70.88** | 66.17 | 61.56 | 61.51 | 45.38 | **75.96** | 51.90 | 65.90 | **68.76** | 66.09 | 60.76 |
| MMTTA* | MM-TTA | 1.99 | 1.99 | 1.99 | 1.99 | 1.99 | 1.99 | 1.99 | 1.99 | 1.99 | 1.99 | 1.99 | 1.99 | 1.99 | 23.88 | | 3.45 |
| READ | MM-TTA | 2.03 | 2.03 | 2.03 | 2.03 | 2.03 | 2.03 | 2.03 | 2.03 | 2.03 | 2.03 | 2.03 | 2.03 | 2.19 | 2.93 | 22.95 | 3.50 |
| MDAA | MM-CTTA | 67.32 | 67.48 | 67.76 | 68.98 | 67.60 | 69.59 | 68.49 | 66.46 | 66.18 | 63.14 | 72.87 | 59.33 | 66.59 | 67.64 | 65.25 | 66.98 |

CTTA methods of CoTTA (Wang et al., 2022) and EATA (Niu et al., 2022); and MM-TTA methods of MMTTA (Shin et al., 2022) and READ (Xiong et al., 2024). To ensure a fair comparison, all methods are based on the pre-trained ViT-baesd CAV-MAE (Gong et al., 2022b) as the multi-modal encoders. When reproduce methods that update the BN layers, we instead update the LN layers to suit the ViT structure. Additionally, we evaluate the Source model as a strong baseline, which is only trained on the source dataset and remains frozen during test time.

Performances of different methods on **progressive single-modality corruption** are listed in Table 1, 2 and 3, where Table 1 is based on the audio corruption, Table 2 and 3are performance on the video corruption in Kinetics50-C and VGGSound-C. The proposed MDAA achieves SOTA performance on most target domains. It is noteworthy that nearly all comparison models collapse and perform significantly worse than non-updating models on VGGSound-C. In contrast, EATA achieves better results by restricting model parameter updates, which facilitates successful adaptation. Specifically, MDAA outperforms previous methods by 3.00%-3.57% and 3.03%-6.22% on average for audio and video tasks in Kinetics50-C, and by 0.94%-1.10% and 0.13%-0.18% on average for audio and video tasks in VGGSound-C. Furthermore, MDAA consistently maintains its lead over other methods in the later stages of the tasks. These results demonstrate that MDAA is more robust against catastrophic forgetting in MM-CTTA. The comparison results of **interleaved modality corruption** tasks are shown in Table 4 and 5. In this task, EATA, which is more good at memorisation, is not dominant in the task of highlighting reliability bias. READ, which is specifically designed to address intra-modal reliability bias, demonstrates strong performance in this area. However, its effectiveness is limited to video corruption in MM-CTTA, as performance drops significantly during audio corruption. In contrast, MDAA is well-adapted to the corruption of different modalities,

outperforming READ by 2.39%-6.84% and EATA by 0.60%-7.28% on average across Kinetics50-C and VGGSound-C.

Table 3: Comparison with SOTA methods on **video progressive single-modality corruption** task in terms of classification Top-1 accuracy (%), with dataset **VGGSound-C** in severity level 5.

| Method | Type | Gauss. | Shot. | Impul. | Defoc. | Glass. | Motion. | Zoom. | Snow | Frost | Fog | Bright. | Cont. | Elastic. | Pixel. | Jpeg | Avg. |
|--------|------|--------|-------|--------|--------|--------|---------|-------|------|-------|-----|---------|-------|----------|--------|------|------|
| | | | | | | | $t$ ———————————————→ | | | | | | | | | | |
| Source | - | 53.02 | 52.90 | 52.98 | **57.20** | **57.38** | 58.37 | 57.48 | 56.40 | 56.46 | 55.41 | **59.16** | 53.73 | 57.22 | 56.44 | 57.33 | 56.10 |
| TENT* | TTA | 51.48 | 50.70 | 50.87 | 51.15 | 50.90 | 51.09 | 50.82 | 50.65 | 50.75 | 50.73 | 50.73 | 50.58 | 50.70 | 50.73 | 50.70 | 50.84 |
| SAR | TTA | 43.12 | 38.99 | 37.77 | 42.43 | 43.84 | 43.61 | 43.79 | 42.13 | 41.26 | 42.83 | 43.84 | 39.34 | 42.75 | 43.49 | 40.38 | 41.97 |
| CoTTA | CTTA | 31.20 | 6.93 | 0.51 | 0.38 | 1.26 | 0.95 | 0.45 | 0.33 | 0.33 | 0.33 | 0.27 | 0.32 | 0.34 | 0.36 | 0.40 | 2.96 |
| EATA | CTTA | 53.57 | 53.70 | 53.57 | 57.00 | 57.29 | **58.46** | **57.77** | 56.24 | **56.57** | 55.45 | 59.06 | 54.13 | **58.24** | **57.22** | 57.38 | 56.38 |
| MMTTA* | MM-TTA | 0.46 | 0.34 | 0.34 | 0.34 | 0.34 | 0.34 | 0.34 | 0.34 | 0.34 | 0.34 | 0.34 | 0.34 | 0.34 | 0.34 | 0.34 | 0.35 |
| READ | MM-TTA | 33.02 | 0.33 | 0.33 | 0.41 | 0.33 | 0.33 | 0.33 | 0.33 | 0.33 | 0.33 | 0.33 | 0.33 | 0.33 | 0.33 | 0.33 | 2.51 |
| MDAA | MM-CTTA | **55.13** | **55.29** | **55.30** | 56.91 | 57.20 | 57.78 | 57.32 | **56.52** | 56.25 | **56.14** | 58.11 | **55.32** | 57.06 | 56.27 | **57.39** | **56.53** |
| | | | | | | | $t$ ←——————————————— | | | | | | | | | | |
| TENT* | TTA | 52.68 | 52.74 | 52.49 | 53.45 | 53.45 | 53.50 | 53.10 | 53.35 | 53.67 | 52.83 | 55.86 | 51.82 | 56.81 | **57.46** | 57.43 | 54.04 |
| SAR | TTA | 40.31 | 39.24 | 38.33 | 41.53 | 41.36 | 44.43 | 43.59 | 42.46 | 41.11 | 41.52 | 42.24 | 38.97 | 43.49 | 40.97 | 46.22 | 41.72 |
| CoTTA | CTTA | 0.33 | 0.33 | 0.35 | 0.33 | 0.33 | 0.33 | 0.32 | 0.46 | 0.36 | 0.35 | 0.56 | 2.24 | 8.36 | 19.22 | 38.95 | 4.85 |
| EATA | CTTA | 53.63 | 53.60 | 53.61 | 57.06 | 57.27 | **58.35** | **57.83** | 56.22 | **56.74** | 55.73 | **59.16** | 54.09 | **58.19** | 57.27 | 57.35 | 56.41 |
| MMTTA* | MM-TTA | 0.34 | 0.34 | 0.34 | 0.34 | 0.34 | 0.34 | 0.34 | 0.34 | 0.34 | 0.34 | 0.34 | 0.34 | 0.34 | 0.34 | 8.32 | 0.87 |
| READ | MM-TTA | 0.32 | 0.32 | 0.32 | 0.32 | 0.32 | 0.32 | 0.32 | 0.32 | 0.89 | 16.32 | 25.48 | 26.95 | 32.93 | 40.25 | 50.31 | 13.05 |
| MDAA | MM-CTTA | **55.30** | **55.38** | **55.25** | 56.90 | 57.19 | 57.79 | 57.32 | **56.50** | 56.31 | **56.22** | 58.13 | **55.28** | 57.07 | 56.30 | **57.47** | **56.56** |

Table 4: Comparison with SOTA methods on **interleaved modality corruption** task in terms of classification Top-1 accuracy (%), with dataset **Kinetics50-C** in severity level 5. A-C and V-C indicates the corrupted modality in current phase.

| Method | V-C Gauss. | A-C Shot. | V-C Gauss. | A-C Impul. | V-C Defoc. | A-C Traff. | V-C Glass. | A-C Motion. | V-C Crowd | A-C Zoom. | V-C Snow | A-C Frost | V-C Rain | A-C Fog | V-C Bright. | A-C Thund. | V-C Cont. | A-C Elastic. | V-C Wind | A-C Pixel. | V-C Jpeg | Avg. |
|--------|------|------|------|------|------|------|------|------|------|------|------|------|------|------|------|------|------|------|------|------|------|------|
| | | | | | | | | | $t$ ——————————→ | | | | | | | | | | | | | |
| Source | 48.71 | 49.98 | 74.03 | 48.98 | **67.69** | 67.89 | 61.82 | 70.92 | 70.29 | 66.14 | 61.36 | 61.35 | 68.02 | 45.34 | 75.94 | 65.20 | 51.82 | 65.84 | 70.38 | 68.73 | 66.11 | 63.17 |
| TENT* | 48.77 | 48.34 | 74.11 | 46.38 | 62.83 | 67.22 | 62.30 | 68.35 | 69.32 | 64.61 | 54.64 | 57.36 | 66.32 | 46.22 | 63.95 | 37.12 | 38.87 | 42.77 | 40.62 | 10.13 | 5.40 | 51.22 |
| SAR | 48.65 | 49.81 | **74.15** | 48.53 | 66.87 | 65.68 | 62.56 | 70.67 | 68.00 | 66.45 | 58.80 | 60.42 | 69.96 | 47.69 | 75.19 | 67.89 | 50.96 | 65.54 | 70.03 | 66.77 | 63.67 | 62.78 |
| CoTTA | 50.21 | 47.72 | 72.16 | 44.96 | 58.80 | 55.53 | 58.28 | 61.35 | 61.52 | 59.73 | 50.15 | 53.92 | 63.15 | 49.48 | 65.16 | 56.71 | 43.97 | 55.47 | 49.68 | 54.79 | 60.14 | 55.85 |
| EATA | 48.81 | 49.70 | 74.07 | 48.96 | 67.75 | 65.35 | 61.96 | **70.95** | 67.98 | 66.05 | 61.60 | 61.49 | 70.40 | 45.29 | **76.11** | 68.13 | 51.85 | 65.98 | 70.51 | 68.80 | 66.15 | 63.23 |
| MMTTA* | 48.63 | 49.20 | 56.24 | 47.79 | 47.52 | 4.76 | 42.96 | 29.87 | 1.96 | 4.13 | 2.27 | 1.96 | 1.96 | 1.92 | 2.03 | 1.96 | 1.92 | 1.96 | 1.92 | 1.99 | 16.90 |
| READ | 51.18 | 53.62 | 73.88 | 54.73 | 68.67 | 67.95 | **67.07** | 70.14 | 68.84 | **67.62** | 62.68 | 64.90 | 68.24 | 59.46 | 71.43 | 68.16 | 52.60 | 66.35 | 66.18 | 62.85 | 63.35 | 64.28 |
| MDAA | 55.04 | 55.91 | 73.64 | 55.89 | 63.78 | **73.12** | 63.54 | 67.62 | **74.90** | 67.00 | 65.60 | 67.21 | 75.44 | 64.89 | 72.69 | **76.94** | 65.45 | 69.21 | 76.95 | 70.94 | 71.16 | **67.95** |
| | | | | | | | | | $t$ ←—————————— | | | | | | | | | | | | | |
| Source | 48.73 | 49.75 | 74.02 | 48.99 | 67.61 | 65.21 | 61.93 | 70.87 | 67.87 | 66.17 | 61.36 | 61.43 | 70.27 | 45.32 | 75.88 | 67.97 | 51.84 | 65.74 | 70.47 | 68.74 | 66.10 | 63.16 |
| TENT* | 6.72 | 10.03 | 72.92 | 23.89 | 53.44 | 63.96 | 60.40 | 64.98 | 67.75 | 62.93 | 60.28 | 61.20 | 67.26 | 52.92 | 71.88 | 58.44 | 51.24 | 67.56 | 70.84 | 68.87 | 66.41 | 55.40 |
| SAR | 47.97 | 48.73 | 72.21 | 48.46 | 66.39 | 66.75 | 63.69 | 70.28 | 68.00 | 66.33 | 59.24 | 59.85 | 70.17 | 46.42 | 75.52 | 67.59 | 51.12 | 65.23 | 70.24 | 68.31 | 66.18 | 62.79 |
| CoTTA | 56.40 | 58.83 | 71.10 | 58.56 | 65.11 | 64.63 | 65.33 | 67.54 | 64.49 | 65.16 | 62.44 | 61.61 | 68.97 | 60.51 | 73.39 | 68.37 | 49.00 | 66.00 | 67.99 | 66.45 | 65.62 | 63.78 |
| EATA | 48.76 | 49.76 | 73.89 | 48.86 | 67.75 | 65.20 | 61.89 | 70.67 | 68.28 | 66.16 | 61.64 | 61.44 | 70.33 | 45.57 | **75.96** | 68.30 | 51.88 | 65.87 | 70.51 | 68.59 | 65.98 | 63.20 |
| MMTTA* | 1.96 | 1.96 | 2.00 | 1.96 | 1.96 | 2.00 | 2.00 | 2.03 | 1.96 | 2.03 | 1.96 | 2.00 | 1.96 | 2.52 | 18.31 | 4.88 | 43.11 | 49.71 | 62.28 | 55.69 | 60.04 | 15.35 |
| READ | 51.49 | 52.29 | 71.09 | 50.70 | 62.44 | 63.72 | 65.01 | 66.42 | 66.39 | 64.86 | 60.97 | 64.47 | 67.85 | 62.48 | 74.68 | 72.12 | 54.12 | 69.15 | 70.27 | **69.74** | **68.37** | 64.22 |
| MDAA | **70.44** | **70.21** | **77.35** | **70.12** | **72.13** | **76.56** | **70.05** | **72.44** | **76.07** | **71.46** | **68.94** | **68.72** | **75.92** | **66.51** | 73.32 | **75.10** | **61.78** | **68.05** | **73.52** | 68.56 | 64.91 | **71.06** |

Table 5: Comparison with SOTA methods on **interleaved modality corruption** task in terms of classification Top-1 accuracy (%), with dataset **VGGSound-C** in severity level 5.

| Method | V-C Gauss. | A-C Shot. | V-C Gauss. | A-C Impul. | V-C Defoc. | A-C Traff. | V-C Glass. | A-C Motion. | V-C Crowd | A-C Zoom. | V-C Snow | A-C Frost | V-C Rain | A-C Fog | V-C Bright. | A-C Thund. | V-C Cont. | A-C Elastic. | V-C Wind | A-C Pixel. | V-C Jpeg | Avg. |
|--------|------|------|------|------|------|------|------|------|------|------|------|------|------|------|------|------|------|------|------|------|------|------|
| | | | | | | | | | $t$ ——————————→ | | | | | | | | | | | | | |
| Source | 53.05 | 52.91 | 37.32 | 52.98 | 57.19 | 21.24 | 57.37 | 58.37 | 16.89 | 57.45 | 56.37 | 56.47 | 21.82 | 55.41 | 59.19 | 27.37 | 53.75 | 57.19 | 25.66 | 56.44 | 57.33 | 47.23 |
| TENT* | 53.19 | 52.80 | 3.43 | 50.52 | 53.15 | 0.65 | 51.83 | 53.10 | 0.60 | 52.64 | 50.91 | 51.89 | 0.67 | 51.15 | 51.73 | 2.13 | 48.68 | 50.76 | 0.79 | 50.38 | 50.22 | 37.20 |
| SAR | 53.16 | 53.33 | 34.26 | 53.17 | 56.94 | 11.27 | 57.14 | 58.29 | 9.30 | 57.65 | 56.11 | 56.78 | 13.36 | 55.94 | 57.87 | 17.95 | 53.13 | 56.51 | 20.46 | 55.30 | 55.79 | 44.94 |
| CoTTA | 52.24 | 52.26 | 10.66 | 49.15 | 49.22 | 1.68 | 46.15 | 46.38 | 1.34 | 44.84 | 44.66 | 44.13 | 0.65 | 43.57 | 43.23 | 4.83 | 35.16 | 34.85 | 0.56 | 36.10 | 35.34 | 32.24 |
| EATA | 53.63 | 53.70 | **40.51** | 53.59 | 57.21 | 30.81 | 57.45 | 58.49 | 29.80 | 57.85 | 56.37 | 56.85 | 30.55 | 56.72 | 59.13 | **37.29** | 54.31 | 58.27 | 32.58 | 57.28 | 57.55 | 50.00 |
| MMTTA* | 8.27 | 0.34 | 0.34 | 0.34 | 0.34 | 0.34 | 0.34 | 0.34 | 0.34 | 0.34 | 0.34 | 0.34 | 0.34 | 0.34 | 0.34 | 0.34 | 0.34 | 0.34 | 0.34 | 0.34 | 0.34 | 0.72 |
| READ | 53.78 | 53.91 | 39.83 | 54.17 | **57.81** | 26.00 | **58.14** | **59.42** | 21.63 | **59.03** | **57.38** | **58.29** | 22.79 | **57.71** | **59.32** | 26.07 | **55.46** | **58.36** | 18.29 | **57.36** | **57.63** | 48.21 |
| MDAA | **55.09** | **55.31** | 38.60 | **55.31** | 56.89 | **34.83** | 57.20 | 57.69 | **34.65** | 57.35 | 56.47 | 56.27 | **34.28** | 56.17 | 58.05 | 36.86 | 55.33 | 57.00 | **35.53** | 56.28 | 57.35 | **50.60** |
| | | | | | | | | | $t$ ←—————————— | | | | | | | | | | | | | |
| Source | 53.01 | 52.88 | 37.31 | 52.97 | 57.20 | 21.25 | 57.42 | 58.41 | 16.89 | 57.49 | 56.37 | 56.49 | 21.81 | 55.43 | 59.16 | 27.37 | 53.74 | 57.19 | 25.66 | 56.42 | 57.29 | 47.23 |
| TENT* | 24.80 | 42.44 | 1.65 | 49.78 | 51.65 | 0.32 | 51.64 | 51.52 | 0.34 | 51.35 | 50.51 | 50.89 | 0.43 | 51.27 | 52.07 | 1.14 | 51.14 | 54.94 | 1.64 | 56.50 | 56.92 | 35.85 |
| SAR | 51.90 | 51.78 | 25.90 | 51.30 | 54.73 | 5.11 | 54.82 | 56.52 | 7.82 | 56.31 | 54.63 | 55.49 | 13.44 | 55.12 | 57.59 | 15.07 | 53.57 | 56.92 | 15.04 | 56.53 | 57.30 | 43.19 |
| CoTTA | 42.97 | 43.65 | 6.10 | 44.29 | 44.97 | 1.09 | 45.45 | 45.60 | 2.20 | 49.53 | 48.97 | 49.89 | 13.53 | 51.33 | 53.72 | 19.60 | 52.45 | 56.04 | 18.76 | 56.22 | 55.88 | 38.20 |
| EATA | 53.77 | 53.65 | **40.39** | 53.47 | 57.17 | 30.49 | 57.36 | 58.59 | 30.16 | 57.82 | 56.21 | 56.64 | 31.22 | 56.77 | **59.24** | **37.44** | 54.30 | **58.18** | 33.01 | 57.36 | 57.53 | 50.05 |
| MMTTA* | 0.34 | 0.34 | 0.34 | 0.34 | 0.34 | 0.34 | 0.34 | 0.34 | 0.34 | 0.34 | 0.34 | 0.34 | 0.34 | 0.34 | 0.34 | 0.34 | 0.34 | 0.34 | 8.36 | 18.74 | 46.32 | 3.79 |
| READ | 54.35 | 54.56 | 25.03 | 54.38 | **57.99** | 17.67 | **57.76** | **58.74** | 20.57 | **58.66** | **56.91** | **57.58** | 20.81 | **58.04** | 59.10 | 33.82 | **55.54** | 58.13 | 32.75 | **57.80** | **58.34** | 48.03 |
| MDAA | **55.30** | **55.41** | 38.64 | **55.29** | 56.91 | **35.40** | 57.14 | 57.81 | **34.85** | 57.34 | 56.48 | 56.29 | **34.52** | 56.23 | 58.24 | 37.22 | 55.31 | 57.13 | **35.12** | 56.30 | 57.47 | **50.69** |

## 4.3 ABLATION STUDY

In this section, we conduct three ablation studies on both video-corrupted Kinetics50-C dataset and audio-corrupted VGGSound-C dataset in severity level 5, with the batch size of 64. For simplicity, in the following section we use KS-video and VGG-audio to represent these two tasks.

**Component analysis.** To verify the effectiveness of each MDAA component, we adopt an ablation study *w.r.t* three components as shown in Table 6. As observed, the model using only AC under-performs, with an average accuracy 0.5%-0.57% lower than READ on KS-video, and the model

even collapses on VGG-audio, with an average accuracy of 0.47%-0.56%. This occurs because, while AC can prevent model forgetting, it cannot filter out unreliable samples, leading to issues with error accumulation and reliability bias. In contrast, the addition of DSM significantly improves performance, with gains of 1.73%-7.52% on KS-video and 33.74%-33.90% on VGG-audio. The introduction of SPS allows the model to learn from more possibly correct labels at the same time, thus further improving the performances on most tasks.

**Reliable selection threshold.** To examine the effect of the threshold $\theta$ on the DSM, we plot the model's performance with $\theta$ of 0, 1e-4, 5e-4, 1e-3, 2e-3 and 5e-3 in Fig.4(A-B). The performance of the model on both datasets exhibits an increasing and then decreasing trend. When $\theta$ is close to 0, the ACs are updated for nearly every sample, introducing more error. Conversely, when $\theta$ increases too much, the ACs do not learn from new inputs since no samples

Table 6: Ablation studies on different component combinations. Grey denotes the default setting.

| Method | KS-video | | VGG-audio | |
|---|---|---|---|---|
| | $\rightarrow$ | $\leftarrow$ | $\rightarrow$ | $\leftarrow$ |
| READ | 62.32 | 62.59 | 23.93 | 22.39 |
| MDAA (ACs) | 61.82 | 62.02 | 0.47 | 0.56 |
| MDAA (ACs+DSM) | 63.55 | **69.54** | 34.87 | 34.85 |
| MDAA (ACs+DSM+SPS) | **65.43** | 69.30 | **35.82** | **35.77** |

can pass through the DSM, leading to a decline in performance. Therefore, the we choose 1e-3 a moderate valuefor both datasets as the threshold.

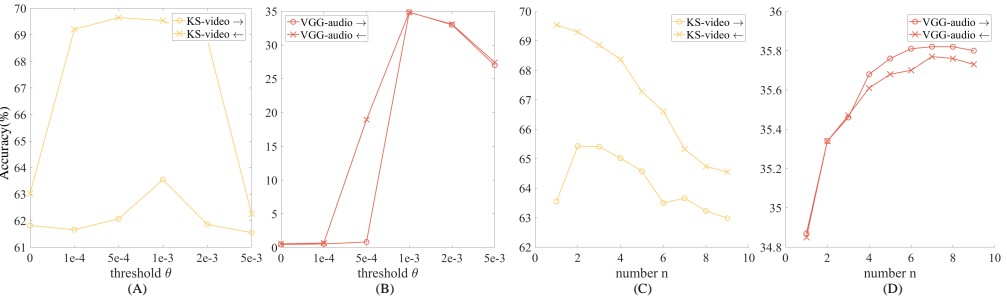

Figure 4: (A-B) Comparison between different threshold $\theta$. (C-D) Comparison of reconstructed pseudo-labels using different numbers of categories.

**Soft label reconstruction.** In this part we determine a suitable number $n$ to reconstruct the pseudo-labels. Given the sorted top $n$ distribution set $C = \{c_1, c_2, \ldots, c_n\}$, we assign weights to each class in a decreasing manner using the formula $\alpha_i = round((n + 1 - i)/\sum_{i=1}^{n} i)$. The results of the model are plotted in Fig.4(C-D) with $n$ chosen from 1 to 9. While using soft labels can inevitably introduce error, there are instances where performance with one-hot labels may exceed that of the SPS (e.g., KS-video $\rightarrow$). However, SPS remains beneficial as the dataset becomes larger and more complex. In fact, performance using SPS surpasses that of one-hot labels in most tasks, making SPS a worthwhile trade-off in MM-CTTA. Generally, the optimal number for reconstructions depends on the number of categories in the dataset, and a larger number of categories warrants a larger $n$. However, more classes will be included in the reconstruction as $n$ increase, thereby introducing more error and reducing model performance. Therefore in SPS, we determine to use top 2 and 7 classes to reconstructed label in Kinetics50-C and VGGSound-C respectively.

## 5 CONCLUSION

In this paper, we analysed the factors that affect the model in the MM-CTTA task (*i.e.,* error accumulation, catastrophic forgetting and reliability bias) and demonstrate that typical TTA methods are not suitable for the MM-CTTA task. To address the impact of these factors, we propose a new paradigm MDAA that introduce analytic learning to TTA for the first time. Instead of just adapting the model to the target domain, MDAA integrates the target domain into source domain, and thus prevent network from forgetting. With the help of DSM and SPS, model is able to dynamically and comprehensively process the information provided by each modality and use reliable samples to update. In the future, we will try to adapt this paradigm to more modalities to solve more challenging problems in real scenarios.

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

## A    PROOFS OF THEOREMS

In this section, we provide comprehensive proofs of **Theorems 1** and **2** presented in the main paper.

*Proof of **Theorem 1***. known the optimal problem in *Eqn.*4 can be further written as:

$$\underset{\mathbf{W}_{\mathrm{S}}}{\operatorname{argmin}} \sum_{k=1}^{N_{\mathrm{S}}} \omega_k (\mathbf{y}_{\mathrm{S},k} - \mathbf{x}_{\mathrm{exf},k}\mathbf{W}_{\mathrm{S}})^\top (\mathbf{y}_{\mathrm{S},k} - \mathbf{x}_{\mathrm{exf},k}\mathbf{W}_{\mathrm{S}}) + \gamma\mathbf{W}_{\mathrm{S}}^\top\mathbf{W}_{\mathrm{S}}$$

$$= \underset{\mathbf{W}_{\mathrm{S}}}{\operatorname{argmin}} \sum_{k=1}^{N_{\mathrm{S}}} \omega_k (\mathbf{W}_{\mathrm{S}}^\top\mathbf{x}_{\mathrm{exf},k}^\top\mathbf{x}_{\mathrm{exf},k}\mathbf{W}_{\mathrm{S}} - \mathbf{y}_{\mathrm{S},k}^\top\mathbf{x}_{\mathrm{exf},k}\mathbf{W}_{\mathrm{S}} - \mathbf{W}_{\mathrm{S}}^\top\mathbf{x}_{\mathrm{exf},k}^\top\mathbf{y}_{\mathrm{S},k} + \mathbf{y}_{\mathrm{S},k}^\top\mathbf{y}_{\mathrm{S},k}) + \gamma\mathbf{W}_{\mathrm{S}}^\top\mathbf{W}_{\mathrm{S}}.$$

Note above equation as $L_1$, derive $L_1$ for $\mathbf{W}_{\mathrm{S}}$ as

$$\frac{\partial L_1}{\partial \mathbf{W}_{\mathrm{S}}} = 2\sum_{k=1}^{N_{\mathrm{S}}} \omega_k (\mathbf{x}_{\mathrm{exf},k}^\top\mathbf{x}_{\mathrm{exf},k}\mathbf{W}_{\mathrm{S}} - \mathbf{x}_{\mathrm{exf},k}^\top\mathbf{y}_{\mathrm{S},k}) + 2\gamma\mathbf{W}_{\mathrm{S}}$$

$$= 2\sum_{k=1}^{N_{\mathrm{S}}} (\tilde{\mathbf{x}}_{\mathrm{exf},k}^\top\tilde{\mathbf{x}}_{\mathrm{exf},k}\mathbf{W}_{\mathrm{S}} - \tilde{\mathbf{x}}_{\mathrm{exf},k}^\top\tilde{\mathbf{y}}_{\mathrm{S},k}) + 2\gamma\mathbf{W}_{\mathrm{S}}$$

$$= 2(\tilde{\mathbf{X}}_{\mathrm{exf},k}^\top\tilde{\mathbf{X}}_{\mathrm{exf},k}\mathbf{W}_{\mathrm{S}} - \tilde{\mathbf{X}}_{\mathrm{exf},k}^\top\tilde{\mathbf{Y}}_{\mathrm{S},k}) + 2\gamma\mathbf{W}_{\mathrm{S}}$$

$$= 0.$$

Therefore $\hat{\mathbf{W}}_{\mathrm{S}} = (\tilde{\mathbf{X}}_{\mathrm{exf},\mathrm{S}}^\top\tilde{\mathbf{X}}_{\mathrm{exf},\mathrm{S}} + \gamma\mathbf{I})^{-1}\tilde{\mathbf{X}}_{\mathrm{exf},\mathrm{S}}^\top\tilde{\mathbf{Y}}_{\mathrm{S}} = \mathbf{P}_{\mathrm{S}}^{-1}\mathbf{Q}_{\mathrm{S}}$.

*Proofs of **Theorem 2***. Similar to the proof of **Theorem 1**, we note the optimal formula as $L_2$ and derive it in terms of $\mathbf{W}_{\mathrm{T},t}$ as

$$\frac{\partial L_2}{\partial \mathbf{W}_{\mathrm{T},t}} = 2(\tilde{\mathbf{X}}_{\mathrm{exf},k}^\top\tilde{\mathbf{X}}_{\mathrm{exf},k}\mathbf{W}_{\mathrm{T},t} + \mathbf{X}_{\mathrm{exf},1:t}^\top\mathbf{X}_{\mathrm{exf},1:t}\mathbf{W}_{\mathrm{T},t} - \tilde{\mathbf{X}}_{\mathrm{exf},k}^\top\tilde{\mathbf{Y}}_{\mathrm{S},k} - \mathbf{X}_{\mathrm{exf},1:t}^\top\bar{\mathbf{Y}}_{\mathrm{T},1:t}) + 2\gamma\mathbf{W}_{\mathrm{T},t}$$

$$= 0.$$

Therefore $\hat{\mathbf{W}}_{\mathrm{T},t} = (\tilde{\mathbf{X}}_{\mathrm{exf},\mathrm{S}}^\top\tilde{\mathbf{X}}_{\mathrm{exf},\mathrm{S}} + \mathbf{X}_{\mathrm{exf},1:t}^\top\mathbf{X}_{\mathrm{exf},1:t} + \gamma\mathbf{I})^{-1}(\tilde{\mathbf{X}}_{\mathrm{exf},\mathrm{S}}^\top\tilde{\mathbf{Y}}_{\mathrm{S}} + \mathbf{X}_{\mathrm{exf},1:t}^\top\bar{\mathbf{Y}}_{\mathrm{T},1:t})$.

# B  PSEUDO-CODE FOR MDAA

In this appendix, we provide the pseudo-code for our MDAA in Algorithm 1. For the pre-trained model, we integrate an individual Analytic Classifier (AC) for each network block, using the source dataset to initialize the classifiers as well as the memory bank. During the inference and adaptation periods, the model reconstructs the output labels for each sample using the Soft Pseudo-label Strategy (SPS) and determines which ACs need to be updated through the Dynamic Selecting Mechanism (DSM).

---

**Algorithm 1** **M**ulti-modality **D**ynamic **A**nalytic **A**daptor (MDAA)

---

**Require:** Source datasets $D_{\text{S}} \sim \{\mathbf{X}_{\text{S}}^a, \mathbf{X}_{\text{S}}^v, \mathbf{Y}_{\text{S}}\}$ and target datasets $D_{\text{T},t} \sim \{\mathbf{X}_{\text{T},t}^a, \mathbf{X}_{\text{T},t}^v\}$, pre-trained network $\Phi_{\text{S}}$.

  1. Training phase:

  (1) integrate AC for each network block in $\Phi_{\text{S}}$ through $Eqn.2$ and 3;

  (2) Determine the parameters of each AC using $D_{\text{S}}$ through $Eqn.6$;

  (3) Initialize the memory bank $B_S$ through $Eqn.7$ and 8.

  2. Inference and Adaptation phase:

  **for** Samples in each batch **do**

    (1) Calculate the output *leader* of each classifier and choose *leader* classifier;

    (2) Reconstruct *leader*'s label through SPS ($Eqn.11$);

    **for** Each *follower* classifier: **do**

      Determine whether to update using DSM ($Eqn.10$);

      **if** needs to be updated **then**

        Update parameters through $Eqn.13$;

        Update memory bank through $Eqn.14$ and 15.

      **end if**

    **end for**

  **end for**

---

## C BENCHMARKS, BACKBONE AND IMPLEMENTATION DETAILS

### C.1 DETAILS ABOUT THE BENCHMARKS

ALL experiments are conducted on the two popular multi-modal datasets Kinetics (Kay et al., 2017) and VGGSound (Chen et al., 2020). Yang et al. (2024) further provides their corrupted visual and audio modality for TTA tasks.

Kinetics50 is a subset of the Kinetics dataset (Kay et al., 2017), consisting of 50 randomly selected classes (Yang et al., 2024). It primarily includes videos that focus on human motion-related activities, with each clip lasting approximately 10 seconds and labeled with a single action class. All videos are sourced from YouTube. The Kinetics50 dataset comprises 29,204 visual-audio pairs for training and 2,466 pairs for testing, with the video modality playing a more prominent role in modality pairing.

VGGSound is a large-scale audio-visual dataset containing short audio clips extracted from YouTube videos (Chen et al., 2020), covering 309 distinct everyday audio events. Each clip has a fixed duration of 10 seconds. For this study, we utilize the 157,602 pairs for training and 14,046 pairs for testing. Compared to Kinetics50, VGGSound includes a wider range of categories, introducing additional complexity to the classification task.

Both datasets' visual and audio modalities were extracted following the method described in Gong et al. (2022b). To systematically explore the distributional shifts across modalities, various corruption types were applied to both visual and audio components. Following Yang et al. (2024), 15 corruption types were introduced for the visual modality, each with five levels of severity for comprehensive evaluation. These corruptions include "gaussian noise", "shot noise", "impulse noise", "defocus blur", "glass blur", "motion blur", "zoom blur", "snow", "frost", "fog", "brightness", "contrast", "elastic transform", "pixelate", and "jpeg compression". Similarly, the audio modality was subjected to 6 different corruptions: "gaussian noise", "traffic noise", "crowd noise", "rain", "thunder", and "wind". The corrupted versions of these benchmarks are referred to as Kinetics50-C and VGGSound-C, respectively. Visualizations of sample corrupted video frames and audio spectrograms are provided in Fig.5 and Fig.6.

### C.2 CAV-MAE BACKBONE

CAV-MAE is employed as the pre-trained model for MM-CTTA in this paper. Its architecture consists of 11 Transformer blocks (known as feature encoder networks) dedicated to modality-specific feature extraction, followed by an additional Transformer block (known as fusion network) responsible for cross modal fusion. For the video input, 10 frames are sampled from each clip, from which a single frame is randomly selected and fed into the Transformer encoder for the visual modality. In the case of the audio input, the original 10-second audio waveform is transformed into a 2D spectrogram before being processed by the Transformer encoder for the audio modality (Gong et al., 2022a).

### C.3 IMPLEMENTATION DETAILS

In the final step of Algorithm 1, we determine the hyperparameters for MDAA. The expansion layer dimension, denoted as $\varphi$, theoretically benefits from larger values. However, an excessively large dimension may introduce a significant number of parameters, increasing the computational load. Given the constraints of our available GPU resources, we set the dimension of $\varphi$ to 8000. The necessity of the parameter $\gamma$ in $Eqn.6$ has been established in Zhuang et al. (2022). The model demonstrates stable performance over a wide range of $\gamma$ values, indicating that as long as $\gamma$ is within a reasonable range, its impact on the model's performance remains minimal. After conducting a sweep over five orders of magnitude ($10^{-3}, 10^{-2}, \ldots, 10^{2}$), we set $\gamma$ to 1 for Kinetics-50 and 10 for VGGSound. As discussed in Sections 4.2 and 4.3, the threshold $\theta$ in DSM is fixed at 0.001 for both datasets, while the parameter $n$ in SPS is set to 2 for Kinetics-50 and 7 for VGGSound. All experiments were conducted on an RTX3090 GPU, with results averaged over three runs.

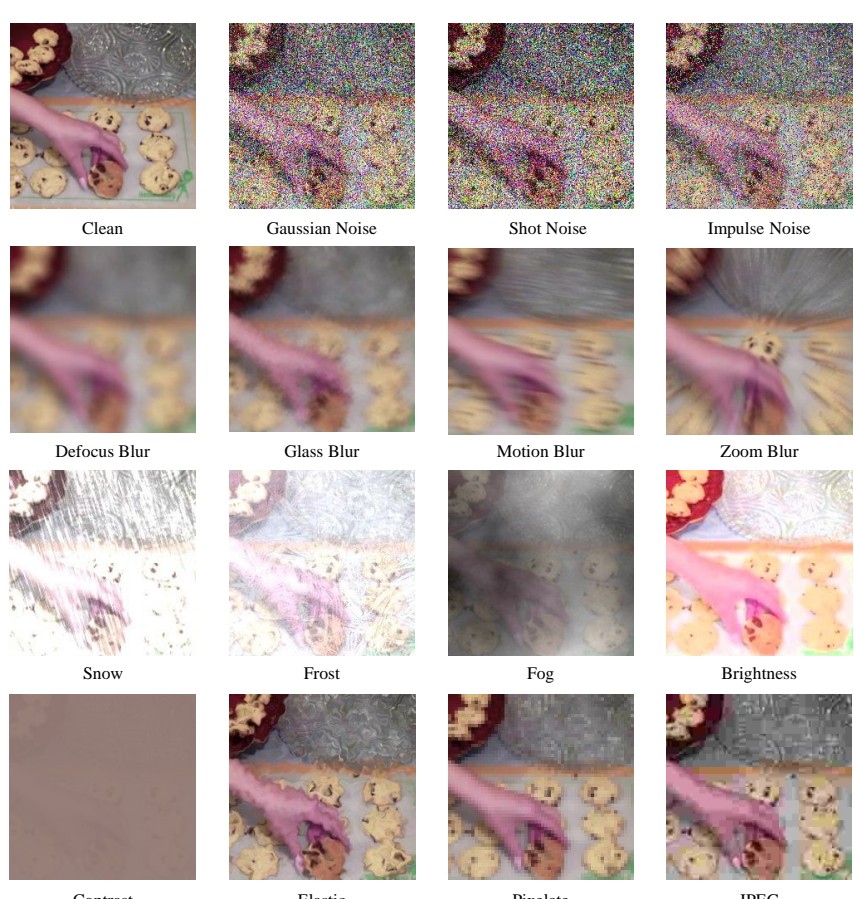

Figure 5: Visualization of 15 corruption types on the sampled video in Kinetics50-C benchmark.

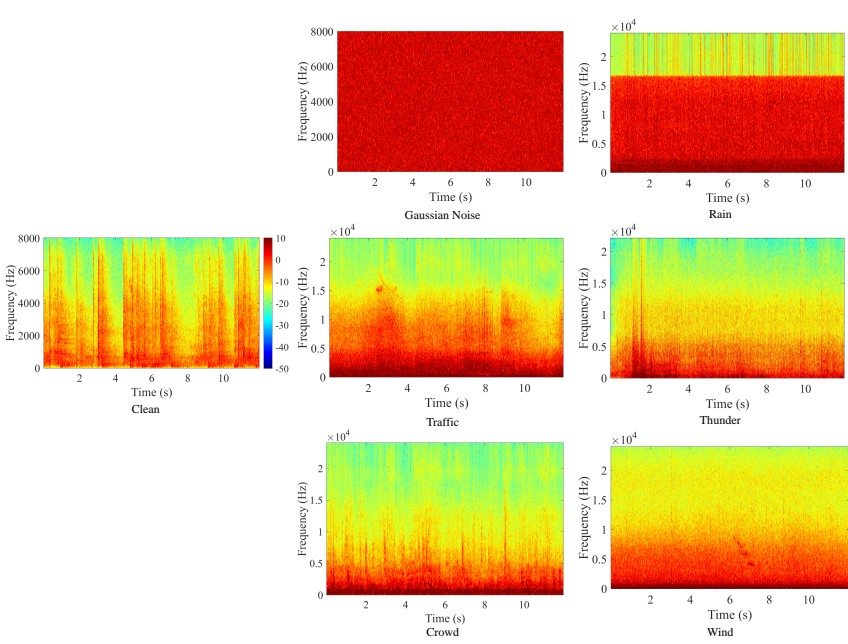

Figure 6: Spectrogram visualization of the clean audio and the corresponding 6 corruption types on the constructed Kinetics50-C benchmark. All Spectrogram use the same range of colorbar.

# D   MORE EXPERIMENT RESULTS

In this appendix we provide four more experiments for reference where Appendix .D.1 and D.2 are the supplementary comparative studies while Appendix .D.3 and D.4 are the further discussion on hyperparameters applied in MDAA.

## D.1   PERFORMANCE COMPARISON ON MULTI-MODAL CORRUPTION

We follow the experimental setup in (Yang et al., 2024) to examine whether our model remains superior to others when the adaptation task is not continual. In this section, the corrupted target domains are treated as independent, and the results for corrupted audio and video modalities are presented in Tables 7, 8, and 9. The results show that the SOTA performance of MDAA is not only due to its ability to combat catastrophic forgetting but also its strong capacity to handle MM-TTA tasks effectively.

Table 7: Comparison with SOTA methods on audio single-modality corruption task in terms of classification Top-1 accuracy (%), using dataset **Kinetics50-C** and **VGGSound-C** in severity level 5. Results of the comparison methods are cite from Yang et al. (2024).

| Method | Type | Kinetics50-C | | | | | | | VGGSound-C | | | | | | |
|--------|------|-------|--------|-------|------|--------|------|------|-------|--------|-------|------|--------|------|------|
| | | Gauss. | Traff. | Crowd | Rain | Thund. | Wind | Avg. | Gauss. | Traff. | Crowd | Rain | Thund. | Wind | Avg. |
| Source | - | 73.7 | 65.5 | 67.9 | 70.3 | 67.9 | 70.3 | 69.3 | 37.0 | 25.5 | 16.8 | 21.6 | 27.3 | 25.5 | 25.6 |
| TENT | TTA | 73.9 | 67.4 | 69.2 | 70.4 | 66.5 | 70.5 | 69.6 | 10.6 | 2.6 | 1.8 | 2.8 | 5.3 | 4.1 | 4.5 |
| SAR | TTA | 73.7 | 65.4 | 68.2 | 69.9 | 67.2 | 70.2 | 69.1 | 37.4 | 9.5 | 11.0 | 12.1 | 26.8 | 23.7 | 20.1 |
| EATA | CTTA | 73.7 | 66.1 | 68.5 | 70.3 | 67.9 | 70.1 | 69.4 | 39.2 | 26.1 | 22.9 | 26.0 | 31.7 | 30.4 | 29.4 |
| MMTTA | MM-TTA | 70.8 | 69.2 | 68.5 | 69.0 | 69.8 | 69.4 | 69.4 | 14.1 | 5.2 | 6.4 | 6.9 | 8.6 | 4.5 | 7.6 |
| READ | MM-TTA | 74.1 | 69.0 | 69.7 | 71.1 | 71.8 | 70.7 | 71.1 | 40.4 | 28.9 | 26.6 | 30.9 | 36.7 | 30.6 | 32.4 |
| MDAA | MM-CTTA | 73.8 | 70.3 | 71.0 | 70.9 | 72.8 | 71.4 | 71.7 | 38.6 | 34.9 | 34.6 | 34.3 | 37.4 | 35.2 | 35.8 |

Table 8: Comparison with SOTA methods on **video** single-modality corruption task in terms of classification Top-1 accuracy (%), with dataset **Kinetics50-C** in severity level 5. Results of the comparison methods are cite from Yang et al. (2024).

| Method | Type | Gauss. | Shot. | Impul. | Defoc. | Glass. | Motion. | Zoom. | Snow | Frost | Fog | Bright. | Cont. | Elastic. | Pixel. | Jpeg | Avg. |
|--------|------|--------|-------|--------|--------|--------|---------|-------|------|-------|------|---------|-------|----------|--------|------|------|
| Source | - | 46.8 | 48.0 | 46.9 | 67.5 | 62.2 | 70.8 | 66.7 | 61.6 | 60.3 | 46.7 | 75.2 | 52.1 | 65.7 | 66.5 | 61.9 | 59.9 |
| TENT | TTA | 46.3 | 47.0 | 46.3 | 67.2 | 62.5 | 71.0 | 67.6 | 63.1 | 61.1 | 34.9 | 75.4 | 51.6 | 66.8 | 67.2 | 62.7 | 59.4 |
| SAR | TTA | 46.7 | 47.4 | 46.8 | 67.0 | 61.9 | 70.4 | 66.4 | 61.8 | 60.6 | 46.0 | 75.2 | 52.1 | 65.7 | 66.4 | 62.0 | 59.8 |
| EATA | CTTA | 46.8 | 47.6 | 47.1 | 67.2 | 62.7 | 70.6 | 67.2 | 62.3 | 60.9 | 46.7 | 75.2 | 52.4 | 65.9 | 66.8 | 62.5 | 60.1 |
| MMTTA | MM-TTA | 46.2 | 46.6 | 46.1 | 58.8 | 55.7 | 62.6 | 58.7 | 52.6 | 54.4 | 48.5 | 69.1 | 49.3 | 57.6 | 56.4 | 54.6 | 54.5 |
| READ | MM-TTA | 49.4 | 49.7 | 49.0 | 68.0 | 65.1 | 71.2 | 69.0 | 64.5 | 64.4 | 57.4 | 75.5 | 53.6 | 68.3 | 68.0 | 65.1 | 62.5 |
| MDAA | MM-CTTA | 55.1 | 55.3 | 55.7 | 64.5 | 62.3 | 67.7 | 65.0 | 61.6 | 63.6 | 57.9 | 72.2 | 54.8 | 66.6 | 67.0 | 65.2 | 62.3 |

Table 9: Comparison with SOTA methods on **video** single-modality corruption task in terms of classification Top-1 accuracy (%), with dataset **VGGSound-C** in severity level 5. Results of the comparison methods are cite from Yang et al. (2024).

| Method | Type | Gauss. | Shot. | Impul. | Defoc. | Glass. | Motion. | Zoom. | Snow | Frost | Fog | Bright. | Cont. | Elastic. | Pixel. | Jpeg | Avg. |
|--------|------|--------|-------|--------|--------|--------|---------|-------|------|-------|------|---------|-------|----------|--------|------|------|
| Source | - | 52.8 | 52.7 | 52.7 | 57.2 | 57.2 | 58.7 | 57.6 | 56.4 | 56.6 | 55.6 | 58.9 | 53.7 | 56.9 | 55.8 | 56.9 | 56.0 |
| TENT | TTA | 52.7 | 52.7 | 52.7 | 56.7 | 56.5 | 57.9 | 57.2 | 55.9 | 56.3 | 56.3 | 58.4 | 54.0 | 57.4 | 56.2 | 56.7 | 55.8 |
| SAR | TTA | 52.9 | 52.8 | 52.9 | 57.2 | 57.1 | 58.6 | 57.6 | 56.3 | 56.7 | 55.9 | 58.9 | 54.0 | 57.0 | 56.0 | 57.0 | 56.1 |
| EATA | CTTA | 53.0 | 52.8 | 53.0 | 57.2 | 57.1 | 58.6 | 57.8 | 56.3 | 56.8 | 56.4 | 59.0 | 54.1 | 57.4 | 56.1 | 57.0 | 56.2 |
| MMTTA | MM-TTA | 7.1 | 7.3 | 7.3 | 44.8 | 41.5 | 48.0 | 45.5 | 27.4 | 23.5 | 30.5 | 46.9 | 24.2 | 40.3 | 40.7 | 45.7 | 32.0 |
| READ | MM-TTA | 53.6 | 53.6 | 53.5 | 57.9 | 57.7 | 59.4 | 58.8 | 57.2 | 57.8 | 55.0 | 59.9 | 55.2 | 58.6 | 57.1 | 57.9 | 56.9 |
| MDAA | MM-CTTA | **54.89** | **55.25** | **55.32** | 63.89 | **62.49** | 67.26 | 65.86 | **64.32** | **65.31** | **61.86** | 73.20 | **61.60** | 67.83 | 69.22 | 68.69 | **63.80** |

## D.2   PERFORMANCE COMPARISON ON SINGLE-MODALITY CONTINUAL CORRUPTION

In this section, we compare the performance of each method under single-modality corruption. This task is similar to the **progressive single-modality corruption** task described in the main text, but here we use a batch size of 64.

Table 10: Comparison with SOTA methods on **audio** single-modality continual corruption task in terms of classification Top-1 accuracy (%), using dataset **Kinetics50-C** and **VGGSound-C** in severity level 5. The best results for each domain are highlighted in **bold**. * means we revise the method from BN to LN for fair comparison.

| Method | Type | Kinetics50-C | | | | | | | VGGSound-C | | | | | | |
|---|---|---|---|---|---|---|---|---|---|---|---|---|---|---|---|
| | | Gauss. | Traff. | Crowd | Rain | Thund. | Wind | Avg. | Gauss. | Traff. | Crowd | Rain | Thund. | Wind | Avg. |
| | | $t$ ———————→ | | | | | | | $t$ ———————→ | | | | | | |
| Source | - | 73.97 | 65.21 | 67.79 | 70.27 | 67.98 | 70.45 | 69.28 | 37.32 | 21.24 | 16.89 | 21.82 | 27.37 | 25.66 | 25.05 |
| TENT* | TTA | 74.44 | 68.04 | 71.30 | 70.25 | 72.53 | 70.35 | 71.15 | 10.76 | 1.15 | 0.40 | 0.32 | 0.51 | 0.31 | 2.24 |
| SAR | TTA | 73.88 | 65.68 | 68.00 | 70.91 | 69.07 | 70.45 | 69.66 | 37.39 | 8.57 | 7.03 | 12.58 | 10.77 | 13.71 | 15.01 |
| CoTTA | CTTA | 73.89 | 66.84 | 68.08 | 67.53 | 71.10 | 69.33 | 69.46 | 39.70 | 34.88 | 35.54 | 33.67 | 39.42 | 33.83 | 36.17 |
| EATA | CTTA | 73.95 | 65.26 | 68.03 | 70.45 | 68.17 | 70.48 | 69.39 | 40.49 | 31.07 | 31.98 | 31.40 | 38.26 | 33.84 | 34.51 |
| MMTTA* | MM-TTA | 69.32 | 69.01 | 69.00 | 69.07 | 68.96 | 68.95 | 69.05 | 14.40 | 1.92 | 0.84 | 0.36 | 0.47 | 0.31 | 3.05 |
| READ | MM-TTA | 74.74 | 68.88 | 70.43 | 70.69 | 72.31 | 69.73 | 71.13 | 40.51 | 25.39 | 20.38 | 20.06 | 21.14 | 16.07 | 23.93 |
| MDAA | MM-CTTA | 73.33 | 71.99 | 73.36 | 73.26 | 74.24 | 73.76 | 73.32 | 38.57 | 34.57 | 34.37 | 34.30 | 37.40 | 35.70 | 35.82 |
| | | $t$ ←——————— | | | | | | | $t$ ←——————— | | | | | | |
| Source | - | 74.00 | 65.20 | 67.92 | 70.24 | 68.01 | 70.43 | 69.30 | 37.31 | 21.25 | 16.89 | 21.81 | 27.37 | 25.66 | 25.05 |
| TENT* | TTA | 73.31 | 70.00 | 71.57 | 70.30 | 68.87 | 71.09 | 70.86 | 0.37 | 0.30 | 0.30 | 0.31 | 0.97 | 3.73 | 1.00 |
| SAR | TTA | 73.29 | 66.79 | 68.14 | 70.71 | 67.97 | 70.36 | 69.55 | 32.02 | 9.92 | 7.04 | 9.49 | 11.41 | 16.03 | 14.32 |
| CoTTA | CTTA | 69.88 | 67.67 | 67.84 | 68.54 | 69.63 | 70.76 | 69.05 | 25.06 | 25.08 | 28.01 | 33.22 | 37.90 | 29.67 | 29.82 |
| EATA | CTTA | 73.96 | 65.30 | 68.15 | 70.36 | 68.18 | 70.40 | 69.39 | 40.65 | 32.32 | 30.65 | 32.23 | 38.16 | 32.39 | 34.40 |
| MMTTA* | MM-TTA | 68.92 | 69.47 | 70.50 | 69.59 | 69.62 | 69.63 | 69.62 | 0.35 | 0.32 | 0.30 | 0.23 | 0.67 | 3.25 | 0.85 |
| READ | MM-TTA | 73.11 | 70.16 | 69.68 | 70.90 | 72.07 | 70.73 | 71.11 | 24.23 | 16.23 | 16.84 | 17.65 | 29.12 | 30.30 | 22.39 |
| MDAA | MM-CTTA | 72.55 | 73.67 | 72.98 | 73.04 | 73.18 | 71.55 | 72.83 | 38.59 | 35.35 | 34.50 | 34.31 | 36.94 | 34.91 | 35.77 |

Table 11: Comparison with SOTA methods on **video** single-modality continual corruption task in terms of classification Top-1 accuracy (%), with dataset **Kinetics50-C** in severity level 5.

| Method | Type | Gauss. | Shot. | Impul. | Defoc. | Glass. | Motion. | Zoom. | Snow | Frost | Fog | Bright. | Cont. | Elastic. | Pixel. | Jpeg | Avg. |
|---|---|---|---|---|---|---|---|---|---|---|---|---|---|---|---|---|---|
| | | $t$ ———————→ | | | | | | | | | | | | | | | |
| Source | - | 48.67 | 49.81 | 49.01 | 67.77 | 61.88 | 70.95 | 66.19 | 61.39 | 61.42 | 45.35 | 75.94 | 51.86 | 65.81 | 68.77 | 66.10 | 60.73 |
| TENT* | TTA | 48.53 | 48.65 | 46.06 | 62.91 | 62.15 | 65.77 | 63.83 | 54.78 | 54.64 | 33.79 | 36.79 | 19.86 | 11.75 | 3.82 | 3.38 | 41.11 |
| SAR | TTA | 48.48 | 49.87 | 48.71 | 66.92 | 62.56 | 70.58 | 66.77 | 59.25 | 60.50 | 47.19 | 75.34 | 50.77 | 65.23 | 66.91 | 64.01 | 60.21 |
| CoTTA | CTTA | 49.17 | 46.65 | 43.69 | 61.82 | 60.00 | 62.34 | 50.60 | 52.13 | 53.35 | | 60.25 | 50.86 | 60.43 | 58.28 | 62.61 | 55.71 |
| EATA | CTTA | 48.76 | 49.84 | 49.03 | 67.78 | 62.02 | 70.92 | 66.20 | 61.55 | 61.53 | 45.38 | 75.97 | 51.90 | 65.91 | 68.74 | 66.09 | 60.77 |
| MMTTA* | MM-TTA | 48.74 | 49.05 | 48.88 | 49.12 | 48.94 | 48.88 | 48.86 | 48.92 | 48.92 | 48.86 | 49.03 | 48.79 | 48.88 | 48.80 | 48.92 | 48.91 |
| READ | MM-TTA | 51.02 | 53.54 | 54.24 | 68.16 | 66.36 | 68.95 | 67.76 | 62.85 | 64.72 | 59.62 | 70.99 | 53.02 | 67.03 | 63.71 | 62.78 | 62.32 |
| MDAA | MM-CTTA | 55.84 | 55.79 | 55.50 | 64.29 | 63.71 | 68.04 | 68.10 | 66.06 | 68.01 | 65.71 | 72.97 | 66.36 | 70.04 | 70.51 | 70.53 | 65.43 |
| | | $t$ ←——————— | | | | | | | | | | | | | | | |
| Source | - | 48.68 | 49.79 | 48.97 | 67.69 | 61.82 | 70.93 | 66.18 | 61.39 | 61.37 | 45.29 | 75.99 | 51.89 | 65.78 | 68.71 | 66.14 | 60.71 |
| TENT* | TTA | 41.93 | 47.19 | 49.22 | 64.80 | 64.33 | 68.05 | 63.86 | 61.42 | 62.12 | 51.65 | 73.92 | 50.80 | 67.47 | 69.03 | 66.80 | 60.17 |
| SAR | TTA | 48.08 | 49.07 | 48.83 | 66.25 | 63.02 | 70.39 | 66.27 | 59.53 | 60.31 | 46.28 | 75.62 | 51.23 | 65.47 | 68.47 | 66.14 | 60.33 |
| CoTTA | CTTA | 56.89 | 57.26 | 56.92 | 66.39 | 65.16 | 68.11 | 66.52 | 60.23 | 62.46 | 50.36 | 73.47 | 50.02 | 66.04 | 67.85 | 67.06 | 62.31 |
| EATA | CTTA | 48.67 | 49.80 | 49.16 | 67.68 | 62.02 | 70.92 | 66.13 | 61.57 | 61.46 | 45.17 | 76.02 | 51.95 | 65.90 | 68.71 | 66.22 | 60.76 |
| MMTTA* | MM-TTA | 49.82 | 49.90 | 49.85 | 49.94 | 49.86 | 49.92 | 49.89 | 49.97 | 49.94 | 49.79 | 50.54 | 50.01 | 54.33 | 56.20 | 59.99 | 51.33 |
| READ | MM-TTA | 52.00 | 52.28 | 51.02 | 62.43 | 64.47 | 66.04 | 65.38 | 61.39 | 65.10 | 62.96 | 75.14 | 53.59 | 68.89 | 69.95 | 68.20 | 62.59 |
| MDAA | MM-CTTA | 70.55 | 70.43 | 70.08 | 71.81 | 70.30 | 72.19 | 70.88 | 69.38 | 69.35 | 67.48 | 73.16 | 62.07 | 68.63 | 67.98 | 65.21 | 69.30 |

Table 12: Comparison with SOTA methods on **video** single-modality continual corruption task in terms of classification Top-1 accuracy (%), with dataset **VGGSound-C** in severity level 5.

| Method | Type | Gauss. | Shot. | Impul. | Defoc. | Glass. | Motion. | Zoom. | Snow | Frost | Fog | Bright. | Cont. | Elastic. | Pixel. | Jpeg | Avg. |
|---|---|---|---|---|---|---|---|---|---|---|---|---|---|---|---|---|---|
| | | $t$ ———————→ | | | | | | | | | | | | | | | |
| Source | - | 53.05 | 52.91 | 52.98 | 57.19 | 57.37 | 58.37 | 57.45 | 56.37 | 56.47 | 55.41 | 59.19 | 53.75 | 57.19 | 56.44 | 57.33 | 56.10 |
| TENT* | TTA | 53.27 | 52.76 | 52.00 | 54.58 | 54.35 | 55.04 | 54.86 | 52.59 | 52.81 | 53.11 | 53.50 | 50.80 | 53.03 | 52.31 | 52.14 | 53.14 |
| SAR | TTA | 53.14 | 53.29 | 53.21 | 56.95 | 56.91 | 58.41 | 57.47 | 56.10 | 56.78 | 56.34 | 58.35 | 53.98 | 57.27 | 55.93 | 56.48 | 56.04 |
| CoTTA | CTTA | 53.03 | 53.08 | 53.03 | 52.92 | 52.28 | 51.84 | 51.41 | 50.83 | 50.08 | 49.66 | 49.22 | 48.85 | 48.30 | 47.62 | 47.20 | 50.62 |
| EATA | CTTA | 53.74 | 53.67 | 53.67 | 57.20 | 57.26 | 58.53 | 57.93 | 56.67 | 56.23 | | 59.04 | 53.63 | 58.19 | 57.36 | 57.48 | 56.47 |
| READ | MM-TTA | 53.77 | 54.26 | 54.28 | 58.04 | 58.00 | 59.09 | 58.84 | 57.43 | 58.18 | 58.12 | 59.38 | 55.99 | 58.30 | 57.51 | 57.91 | 57.27 |
| MDAA | MM-CTTA | 55.63 | 55.91 | 55.88 | 57.50 | 57.77 | 58.27 | 57.84 | 57.05 | 56.72 | 56.77 | 58.67 | 55.78 | 57.55 | 56.87 | 57.84 | 57.07 |
| | | $t$ ←——————— | | | | | | | | | | | | | | | |
| Source | - | 53.01 | 52.88 | 52.97 | 57.20 | 57.42 | 58.41 | 57.49 | 56.37 | 56.49 | 55.43 | 59.16 | 53.74 | 57.19 | 56.42 | 57.29 | 56.10 |
| TENT* | TTA | 50.48 | 50.50 | 50.50 | 53.09 | 53.19 | 53.70 | 53.18 | 52.31 | 53.47 | 53.57 | 55.40 | 52.22 | 56.23 | 56.74 | 56.99 | 53.44 |
| SAR | TTA | 53.02 | 52.95 | 52.71 | 56.78 | 56.63 | 58.18 | 57.38 | 55.97 | 56.69 | 56.45 | 58.46 | 54.03 | 57.41 | 56.53 | 57.16 | 56.02 |
| CoTTA | CTTA | 48.10 | 48.51 | 49.13 | 49.90 | 50.47 | 50.93 | 51.67 | 52.14 | 53.17 | 53.19 | 55.86 | 53.21 | 58.02 | 56.60 | 56.71 | 52.51 |
| EATA | CTTA | 53.69 | 53.66 | 53.61 | 57.26 | 57.34 | 58.46 | 57.84 | 56.35 | 56.82 | 56.72 | 59.21 | 54.10 | 58.28 | 57.39 | 57.54 | 56.55 |
| READ | MM-TTA | 55.33 | 55.34 | 54.93 | 58.16 | 57.88 | 58.82 | 58.50 | 57.24 | 57.94 | 57.86 | 59.65 | 55.67 | 58.79 | 57.87 | 58.10 | 57.47 |
| MDAA | MM-CTTA | 55.85 | 55.87 | 55.94 | 57.40 | 57.85 | 58.25 | 57.95 | 57.02 | 56.89 | 56.68 | 58.66 | 55.84 | 57.58 | 56.92 | 58.00 | 57.11 |

### D.3  DYNAMIC THRESHOLD UPDATE

The threshold $\theta$ we used in DSM is a fixed number. In this section we attempt to update $\theta$ in a dynamic way during the adaptation. We define the threshold $\theta_t^i$ for classifier $i$ in time $t$ as

$$\theta_t^i = \begin{cases} \theta_{t-1}^i + \lambda\left(d_t^i - d_{t-1}^i\right) & ,\text{if } t > 1 \\ \theta_{ini} & ,\text{if } t = 1 \end{cases}, i = \{a, v, m\} \tag{16}$$

$$d_t^i = \frac{\sum_{k=1}^{N_k}\left(max(P_k^{leader}) - max(P_k^i)\right)}{N_k}, i = \{a, v, m\}, \tag{17}$$

where $\lambda$ is the learning rate, $\theta_{ini}$ is the initial threshold and $N_k$ is the batch size. $d_t^i$ is calculated to reflect the gap between *leader* and *follower* $i$. The original intention of this design is to adjust the size of the threshold according to the change of $d_t$, so as to eliminate the statistical bias of different domains. However, as shown in Table 13, such attempt is not only achieve lower performance while needs more variables to be memorized. So only the fixed threshold is used in the formal method.

Table 13: Ablation studies on parameter $\lambda$.

| $\lambda$ | KS-video | | VGG-audio | |
|---|---|---|---|---|
| | $\rightarrow$ | $\leftarrow$ | $\rightarrow$ | $\leftarrow$ |
| 0 | **63.55** | **69.54** | 34.37 | **34.35** |
| 0.01 | 63.49 | 69.31 | 34.35 | 34.30 |
| 0.05 | 63.36 | 69.43 | 34.38 | 34.31 |
| 0.1 | 63.18 | 69.42 | 34.40 | 34.29 |
| 0.2 | 62.93 | 69.33 | **34.41** | 34.31 |

Table 14: Ablation studies on parameter $\alpha$.

| $\alpha$ | KS-video | |
|---|---|---|
| | $\rightarrow$ | $\leftarrow$ |
| 0.9 | 64.29 | 69.29 |
| 0.8 | 64.88 | 69.29 |
| 0.7 | **65.43** | **69.30** |
| 0.6 | 65.20 | 68.81 |
| 0.5 | 35.37 | 37.37 |

### D.4  LABEL WEIGHT DISTRIBUTION

The weight assignment in SPS follows the formula $\alpha_i = round((n+1-i)/\sum_{i=1}^n i)$. In this section we make a toy experiment to explore whether such assignment is reasonable. We take KS-video as example which use top 2 classes for reconstruction. Table 14 show the results on different weight assignment. It can be seen that the performance peaks at 0.7, corresponding to the assignment in main text. Therefore the assignment of weights in the main text is justified.

