# OpenReview forum: "Analytic Continual Test-Time Adaptation for Multi-Modality Corruption"
_ICLR.cc/2025/Conference — Submitted to ICLR 2025_

### Official Review · Reviewer_2Yku · 2024-10-29

**Soundness:** 3
**Presentation:** 2
**Contribution:** 2
**Rating:** 6
**Confidence:** 4

**Summary:**

This paper introduces Multi-modality Dynamic Analytic Adapter (MDAA), a novel approach to Multi-Modal Continual Test-Time Adaptation (MM-CTTA). MDAA addresses key challenges in dynamic real-world scenarios, including error accumulation, catastrophic forgetting, and reliability bias. It leverages Analytic Classifiers (ACs) to prevent model forgetting, alongside a Dynamic Selection Mechanism (DSM) and a Soft Pseudo-label Strategy (SPS) to effectively integrate and filter multi-modal information. Extensive experiments demonstrate that MDAA outperforms current state-of-the-art methods on MM-CTTA tasks, showcasing its reliability and adaptability in handling modality corruption and dynamic changes.

**Strengths:**

- Effective Mitigation of Reliability Bias introduces a Dynamic Selection Mechanism (DSM) that dynamically prioritizes more reliable modalities while suppressing corrupted ones, a crucial feature for real-world scenarios where data quality and reliability may vary significantly across modalities. This reliability management enhances MDAA's robustness against data corruption, a critical improvement over existing MM-CTTA approaches.

- The use of Analytic Classifiers (ACs) within the MDAA framework enables continuous learning from new data while retaining previously acquired knowledge, ensuring effective long-term adaptability. This is particularly valuable in MM-CTTA tasks, where models must generalize from the source domain while adapting to continuously evolving target domains without catastrophic forgetting.

**Weaknesses:**

- In the proposed MDAA method, how can we ensure the model does not overly rely on a single modality when dealing with highly imbalanced modal data? Is there empirical evidence of MDAA’s effectiveness in extreme cases?

- While MM-CTTA aims to handle real-world dynamic changes, can the model effectively respond to previously unseen modality corruption in practical applications? Are there relevant test results? Additionally, how effective is MDAA in rapidly changing scenarios where modality shifts occur frequently?

- The snow image in Figure 5 appears quite similar to the rain image. Could the authors check if there is a potential issue here?

- Does the MDAA method incur significant computational overhead? For practical applications, what measures can help balance model complexity with real-time adaptability?

- The motivation in the paper may benefit from further clarification. The issues of error accumulation, catastrophic forgetting, and reliability bias appear as a straightforward combination of Continual Test-Time Adaptation (CTTA) and Multi-modal Test-Time Adaptation (MM-TTA), where CTTA tackles error accumulation and forgetting, while MM-TTA focuses on reliability bias.

- In the Introduction, it is stated, “Error accumulation, stemming from incorrect pseudo-labels, can mislead models’ adaptation and potentially lead to collapse.” Does this imply that, if pseudo-labels were not used, error accumulation would not occur? If so, does this suggest that using pseudo-labels may be unnecessary?

**Questions:**

Another potential area for improvement in MDAA could be its robustness to label noise. While MDAA incorporates a Soft Pseudo-label Strategy (SPS) to enhance robustness against label noise, the effectiveness of this strategy in highly noisy environments remains to be fully evaluated. Future research could focus on developing more advanced mechanisms for detecting and mitigating label noise across different modalities.

---

> ### Author Response · Authors · 2024-11-19
>
> We appreciate your advice and questions on the practical application of the model's capabilities., here are the responses:
> ***
> **W1:** how can we ensure the model does not overly rely on a single modality when dealing with highly imbalanced modal data? Is there empirical evidence of MDAA’s effectiveness in extreme cases?
> ***
> We are not entirely certain about the specific meaning of "overly rely on a single modality" and have considered two possible interpretations:
>
> + **Reliability Bias (as discussed in the article and illustrated in Figure 1(C)):**
>
> We have addressed this issue through dedicated experiments. In Tables 4 and 5, we evaluate various methods under interleaved modality corruption tasks, where the corrupted modality switches repeatedly over time, and the model does not know in advance which modality is corrupted.
>
> While some comparison methods perform well during video-corruption phases, their performance degrades significantly when the corrupted modality changes. In contrast, The performance of our model is consistent with that at unimodal corruption.
>
> + **Modality Absence:**
>
> If "overly rely on a single modality" refers to situations where only one modality is available for prediction, our model is still effective. This is because our framework includes three independent classifiers. As long as data from at least one modality is available, its corresponding classifier will produce predictions and adapt itself accordingly.
> ***
> **W2:** can the model effectively respond to previously unseen modality corruption in practical applications? Are there relevant test results? Additionally, how effective is MDAA in rapidly changing scenarios where modality shifts occur frequently?
> ***
>
> We would like to clarify that the model is pre-trained on clean (i.e. non-corrupted) dataset, therefore all corruption types in the experiment results (e.g. Gauss., Traff, et al.) are all unseen to the model. Table 1-5 are all test results that can illustrate model’s effectiveness of continually adapting to unseen modality corruption.
>
> The recursive solution of MDAA ensures a globally optimal solution that incorporates both the source domain and all previously encountered test domains. Unlike gradient-based methods, MDAA does not require a large number of samples or epochs to adapt to new domain shifts. Therefore, the performance of MDAA remains consistent across different modality shifting frequencies.
> ***
> **W3:** The snow image in Figure 5 appears quite similar to the rain image. Could the authors check if there is a potential issue here?
> ***
> We follow the method in [1] to implement corruption, with the corruption of this image specifically defined as snow in [1].
>
> [1] Yang M, Li Y, Zhang C, et al. Test-time Adaptation against Multi-modal Reliability Bias[C]//The Twelfth International Conference on Learning Representations. 2024.
> ***
> **W4:** Does the MDAA method incur significant computational overhead? For practical applications, what measures can help balance model complexity with real-time adaptability?
> ***
> We have conducted additional experiments on runtime and VRAM usage. Since MDAA does not rely on backpropagation, it **requires less computational time and consumes lower memory compared to most existing methods**. While MDAA adapts to test data based on the analytic solution, the computational complexity mainly lies on the matrix inversion, which has a time complexity of $O(N^3)$. Here, N is the feature dimension, which is set as 8000 throughout our experiments. Note that the feature dimension does not increase during CTTA period, thus the complexity of each update keeps consistent.
> ***
> **Table 1:** Comparison of methods in terms of runtime and memory usage.
> | Method   | KS-video  | KS-video | VGG-audio | VGG-audio |
> |----------|-------------------|----------------------|--------------------|-----------------------|
> ||time (s) |  memory (MB) |time (s) | memory (MB) |
> |TENT| 1085 | 37620| 2571| 37622|
> |SAR| 2149 | 37620| 4823| 39220|
> |CoTTA | 2301| 55216| 5283| 55326|
> |EATA| 889| 67286| 2313|38300|
> |MMTTA | 2574| 71732| 5966|68840|
> |READ| 1174| 9176| 2611| 9178|
> |**MDAA** | 1024| 8784| 1682| 8750|

---

> ### Author Response · Authors · 2024-11-19
>
> **W5:** The motivation in the paper may benefit from further clarification.
> ***
> We appreciate the reviewer’s acknowledgment of the challenges in MM-CTTA. MM-CTTA was first introduced in [2], focusing on enabling multimodal networks to adapt to dynamic target domains in an online manner within the context of 3D segmentation for autonomous driving. Building upon this foundation, our work emphasizes addressing the challenges of MM-CTTA in real-world applications.
>
> For instance, we adopt a more challenging setting that combines the difficulties of MM-TTA and CTTA, where the model must handle continuous interleaved modality corruptions. Furthermore, we introduce even more extreme tasks, such as progressive single-modality corruption, where test data is processed in an online manner (i.e., each sample is seen only once). This setting closely aligns with practical applications requiring rapid adaptation, and our method demonstrates significantly faster adaptation compared to existing approaches.
>
> [2] Cao H, Xu Y, Yang J, et al. Multi-modal continual test-time adaptation for 3d semantic segmentation[C]//Proceedings of the IEEE/CVF International Conference on Computer Vision. 2023: 18809-18819.
> ***
> **W6:** Does this imply that, if pseudo-labels were not used, error accumulation would not occur? If so, does this suggest that using pseudo-labels may be unnecessary?
> ***
> Using pseudo-labels for model updates has been widely recognized as an effective approach for generating supervised signals in previous TTA works [3–7]. However, since TTA methods aim to adapt to test data without access to ground truth for supervision, error accumulation is an unavoidable challenge during the continual adaptation process.
>
> In our work, we introduce DSM and SPS modules to minimize errors introduced by pseudo-labels. The effectiveness of these modules is demonstrated through comprehensive ablation studies.
>
> **Table 2:** Ablation studies on different component combinations.
> | Method| KS-video | KS-video | VGG-audio | VGG-audio |
> |---------------------------|----------------------------|---------------------------|----------------------------|---------------------------|
> ||$\xrightarrow{}$ |$\xleftarrow{}$ |$\xrightarrow{}$ | $\xleftarrow{}$ |
> | READ| 62.32 | 62.59| 23.93| 22.39|
> | MDAA (ACs) | 61.82| 62.02| 0.47| 0.56|
> | MDAA (ACs+DSM)| 63.55| **69.54**| 34.87| 34.85|
> | MDAA (ACs+SPS)| 61.33| 67.47| 32.50| 33.00|
> | MDAA (ACs+DSM+SPS)| **65.43**| 69.30| **35.82**| **35.77**|
> ***
> **Q1:** Another potential area for improvement in MDAA could be its robustness to label noise. the effectiveness of this strategy in highly noisy environments remains to be fully evaluated. Future research could focus on developing more advanced mechanisms for detecting and mitigating label noise across different modalities.
> ***
> In MM-CTTA, noise primarily arises from corrupted data, which can lead to incorrect labels during model updates. To address this, we implement modality compensation through DSM, effectively mitigating the risks associated with noise.
>
> For our experiments, we followed the corruption generation process outlined in [3], applying the highest corruption level (Level 5) for each test sample. This level is particularly challenging and has been used as a benchmark in other TTA works [3,5,7].
>
> In future work, we plan to explore more robust filtering mechanisms to further enhance the practicality and resilience of our method in real-world applications.
>
> [3] Yang M, Li Y, Zhang C, et al. Test-time Adaptation against Multi-modal Reliability Bias[C]//The Twelfth International Conference on Learning Representations. 2024.
>
> [4] Wang D, Shelhamer E, Liu S, et al. Tent: Fully Test-Time Adaptation by Entropy Minimization[C]//International Conference on Learning Representations.
>
> [5] Lei, Jixiang, and Franz Pernkopf. "Two-level test-time adaptation in multimodal learning." ICML 2024 Workshop on Foundation Models in the Wild. 2024.
>
> [6] Niu S, Wu J, Zhang Y, et al. Towards Stable Test-time Adaptation in Dynamic Wild World[C]//The Eleventh International Conference on Learning Representations.
>
> [7] Wang Q, Fink O, Van Gool L, et al. Continual test-time domain adaptation[C]//Proceedings of the IEEE/CVF Conference on Computer Vision and Pattern Recognition. 2022: 7201-7211.
> ***
>
> In light of these clarifications, would you consider increasing your score for our paper? Otherwise, could you let us know any additional changes you would like to see in order for this work to be accepted?

---

> > ### Comment · Reviewer_2Yku · 2024-11-27
> >
> > Thank you for your detailed and thoughtful response. The clarifications and additional experimental results effectively address my concerns. The experimental evidence provided, along with the explanations of the proposed methods, strengthens the case for your approach. Based on this, I have decided to raise my score.

---

### Official Review · Reviewer_KbiA · 2024-11-03

**Soundness:** 3
**Presentation:** 2
**Contribution:** 3
**Rating:** 6
**Confidence:** 4

**Summary:**

The paper presents a novel method called Multi-modality Dynamic Analytic Adapter (MDAA) to tackle the challenges of Multi-Modal Continual Test-Time Adaptation (MM-CTTA). In the MM-CTTA setup, pre-trained models are adapted in an online fashion to corrupted, multi-modal data (e.g., audio, video) without requiring access to original, labeled training data. The core idea in MDAA is its use of analytic classifiers - linear classifiers trained through recursive least squares that effectively store autocorrelation matrices in memory. This approach allows MDAA to retain essential knowledge from the original training data while adapting to ongoing data shifts. To further enhance adaptation, the paper introduces a Dynamic Selection Mechanism (DSM) that dynamically identifies and prioritizes reliable samples, ensuring that updates are informed by high-quality data. Additionally, a Soft Pseudo-label Strategy (SPS) refines pseudo-labels by assigning weighted probabilities across the top-k label predictions, rather than relying on one-hot labels, thus improving the model’s robustness to noisy labels.

The paper performs experiments on two datasets, Kinetics50-C and VGGSound-C, across a range of corruption types. MDAA is tested against state-of-the-art methods in both single-modality corruption and interleaved multi-modal corruption scenarios, demonstrating superior adaptability and robustness.

**Strengths:**

1. The majority of work in the continual test-time adaptation area focus on a single modality. While some works exist in the multi-modal space, they do not address the catastrophic forgetting issue well. MDAA introduces a recursive learning scheme which is tailored for tackling this issue. This makes MDAA highly relevant for practical applications.

2. Extensive experiment results on multiple scenarios are provided.

**Weaknesses:**

1. The classifiers in MDAA are fitted using a least squares loss function. It is well-known that models trained with least squares loss can lack robustness to outliers and are often outperformed by models trained with cross-entropy loss. It would be useful to see a comparison of the source model performance when trained with cross-entropy loss versus least squares loss.

2. The formulation restricts adaptation only to the classification layer. It has been shown in [1] how adapting the encoders themselves can improve performance. While [1] requires knowledge of which modality is corrupted, it would be informative to know what is the drop in performance due to this compromise.

3. The choice of leader network in DSM seems arbitrary in the absence of experiments which validate the choice. For example, how does choosing the lowest entropy network work? Furthermore, how what is the frequency of updates using DSM? These questions need to be clarified.

[1] Lei, Jixiang, and Franz Pernkopf. "Two-level test-time adaptation in multimodal learning." ICML 2024 Workshop on Foundation Models in the Wild. 2024.

**Questions:**

1. What accounts for CoTTA’s significantly low performance (~5%) despite its use of a stochastic restoration mechanism? Additionally, do all baseline methods incorporate separate classifiers for each data stream?

2. How does MDAA’s training time compare to baselines that rely on backpropagation, given that MDAA requires computation of large autocorrelation matrices and their inverses?

---

> ### Author Response · Authors · 2024-11-19
>
> **W1:** comparison of the source model performance when trained with cross-entropy loss versus least squares loss
> ***
> Our learning goal is to map features to pseudo-labels using ridge regression rather than optimizing through back-propagation. Therefore, we are constrained to using a linear loss function, specifically the least squares loss, rather than cross-entropy loss.
> ***
> **W2:** The formulation restricts adaptation only to the classification layer. It has been shown in [1] how adapting the encoders themselves can improve performance. While [1] requires knowledge of which modality is corrupted, it would be informative to know what the drop in performance is due to this compromise
> ***
> The work [1] mentioned by the reviewer is an approach that adapts the encoders to achieve TTA. However, such an approach inevitably shift the feature space of entire model, leading to catastrophic forgetting. Consequently, **[1] is not suitable for CTTA tasks as addressed in our work**. Furthermore, [1] requires prior knowledge about which modality is corrupted, making it inapplicable to scenarios involving interleaved modality corruption.
>
> In contrast, our method updates classifiers by solving the ridge regression problem using the least square error. This approach enables the classifiers to retain prior knowledge from both the source and test domains. To preserve the conditions of the ridge regression problem, parameter updates are restricted to the output layer (i.e., the classifier), while all preceding layers remain frozen.
>
> In the future, we aim to explore improved strategies to enhance the encoder's feature extraction capabilities during the CTTA process.
>
> [1] Lei, Jixiang, and Franz Pernkopf. "Two-level test-time adaptation in multimodal learning." ICML 2024 Workshop on Foundation Models in the Wild. 2024.
> ***
> **W3:** The choice of leader network in DSM seems arbitrary in the absence of experiments which validate the choice. For example, how does choosing the lowest entropy network work? Furthermore, how what is the frequency of updates using DSM? These questions need to be clarified.
> ***
> The selection process in DSM is not arbitrary. **The prediction with the highest probability** is chosen through DSM as the pseudo-label (i.e., the leader prediction) for subsequent updates, rather than using the lowest entropy. This approach is based on the observation that higher prediction probabilities generally reflect greater model confidence and, therefore, higher reliability [1–2]. This operation is repeated for each sample within a batch, hence the **update frequency using DSM is once per batch**. We will highlight corresponding context in the revised manuscript.
>
> [1] Boudiaf M, Mueller R, Ben Ayed I, et al. Parameter-free online test-time adaptation[C]//Proceedings of the IEEE/CVF Conference on Computer Vision and Pattern Recognition. 2022: 8344-8353.
>
> [2] Wang Z, Luo Y, Zheng L, et al. In search of lost online test-time adaptation: A survey[J]. International Journal of Computer Vision, 2024: 1-34.

---

> > ### Author Response · Authors · 2024-11-19
> >
> > **Q1**
> > ***
> > What accounts for CoTTA’s significantly low performance (~5%) despite its use of a stochastic restoration mechanism?
> > ***
> > We need to clarify that CoTTA's low performance primarily stems from our task setting. In the progressive single-modality corruption task, each batch contains only one sample. This setup results in more frequent adaptation phases, significantly increasing the severity of the forgetting issue.
> >
> > Additionally, CoTTA's updating strategy differs from most methods. While many approaches update only the normalization layers, CoTTA updates all model parameters [1], which further amplifies the forgetting effect. As CoTTA was originally designed to operate with a batch size of 200 [1], we believe these two discrepancies are the primary reasons for performance decline of CoTTA.
> >
> > To support this perspective, we provide comparative results on CoTTA with batch sizes of 1 and 64. These results demonstrate a significant improvement in CoTTA's performance as batch sizes increase, although its performance remains lower than that of our MDAA.
> >
> > **Table 1:** Comparison of different batch size on Kinetics50-C
> > | Method        | Batch size | Gauss. | Traff. | Crowd | Rain  | Thund. | Wind  | Avg.  |
> > |------------------|------------|--------|--------|-------|-------|--------|-------|-------|
> > | CoTTA →     | 1    | 19.67  | 4.10   | 2.11  | 2.03  | 2.03   | 2.03  | 5.33  |
> > | CoTTA →    | 64    | 73.89  | 66.84  | 68.08 | 67.53 | 71.10  | 69.33 | 69.46 |
> > | MDAA →     | 1    | 72.87  | 71.45  | 72.91 | 72.26 | 73.20  | 73.80 | 72.75 |
> > | MDAA →      | 64  | 73.33  | 71.99  | 73.36 | 73.26 | 74.24  | 73.76 | 73.32 |
> > | CoTTA ←     | 1    | 1.99   | 1.99   | 2.92  | 5.07  | 11.64  | 32.56 | 9.36 |
> > | CoTTA ←     | 64    | 69.88  | 67.67  | 67.84 | 68.54 | 69.63  | 70.76 | 69.05 |
> > | MDAA ←      | 1    | 74.86  | 72.63  | 72.87 | 72.26 | 73.40  | 72.02 | 73.01 |
> > | MDAA ←     | 64   | 72.55  | 73.67  | 72.98 | 73.04 | 73.18  | 71.55 | 72.83 |
> >
> > **Table 2:** Comparison of different batch size on VGGSound-C
> > | Method        | Batch size | Gauss. | Traff. | Crowd | Rain  | Thund. | Wind  | Avg.  |
> > |------------------|------------|--------|--------|-------|-------|--------|-------|-------|
> > | CoTTA →     | 1  |  5.85   | 1.35   | 0.52  | 0.53  | 0.57   | 0.38  | 1.53  |
> > | CoTTA →    | 64   | 39.70  | 34.88  | 35.54 | 33.67 | 39.42  | 33.83 | 36.17 |
> > | MDAA →     | 1   |38.80  | 34.91  | 34.63 | 34.59 | 37.70  | 35.85 | 36.08 |
> > | MDAA →      | 64| 38.57  | 34.57  | 34.37 | 34.30 | 37.40  | 35.70 | 35.82 |
> > | CoTTA ←     | 1  |  0.30   | 0.30   | 0.30  | 0.30  | 0.30   | 0.36  | 0.31  |
> > | CoTTA ←     | 64 | 25.06  | 25.08  | 28.01 | 33.22 | 37.90  | 29.67 | 29.82 |
> > | MDAA ←      | 1  |  38.95  | 35.46  | 34.66 | 34.70 | 37.31  | 35.20 | 36.05 |
> > | MDAA ←     | 64  |  38.59  | 35.35  | 34.50 | 34.31 | 36.94  | 34.91 | 35.77 |
> >
> > [1] Wang Q, Fink O, Van Gool L, et al. Continual test-time domain adaptation[C]//Proceedings of the IEEE/CVF Conference on Computer Vision and Pattern Recognition. 2022: 7201-7211.
> > ***
> > **Q2:** How does MDAA’s training time compare to baselines that rely on backpropagation, given that MDAA requires computation of large autocorrelation matrices and their inverses?
> > ***
> > We have conducted additional experiments on runtime and VRAM usage. Since MDAA does not rely on backpropagation, it **requires less computational time and consumes lower memory compared to most existing methods**. While MDAA adapts to test data based on the analytic solution, the computational complexity mainly lies on the matrix inversion, which has a time complexity of $O(N^3)$. Here, N is the feature dimension, which is set as 8000 throughout our experiments. Note that the feature dimension does not increase during CTTA period, thus the complexity of each update keeps consistent.
> >
> > **Table 3:** Comparison of methods in terms of runtime and memory usage.
> > | Method   | KS-video  | KS-video | VGG-audio | VGG-audio |
> > |----------|-------------------|----------------------|--------------------|-----------------------|
> > |    |time (s) |  memory (MB) |time (s) | memory (MB) |
> > | TENT    | 1085              | 37620                | 2571               | 37622   |
> > | SAR      | 2149              | 37620                | 4823               | 39220   |
> > | CoTTA    | 2301              | 55216                | 5283               | 55326   |
> > | EATA     | 889               | 67286                | 2313               | 38300     |
> > | MMTTA   | 2574              | 71732                | 5966               | 68840    |
> > | READ     | 1174              | 9176                 | 2611               | 9178     |
> > | **MDAA** | 1024              | 8784                 | 1682               | 8750    |
> > ***
> > In light of these clarifications, would you consider increasing your score for our paper? Otherwise, could you let us know any additional changes you would like to see in order for this work to be accepted?

---

> > > ### Comment · Reviewer_KbiA · 2024-11-30
> > >
> > > My point regarding W1 is referring to the potential drop in source model performance when you apply a least square loss. I understand that MDAA requires a least square loss to work, but treating classification problems as regression is not standard practice and is known to decrease performance. If I fit the source model classifiers using cross-entropy and not adapt them, is that source model better than the least squared version?
> > >
> > > For point W2, you are again going back to how your method works. I don't see any definitive proof that feature space adaptation is detrimental for CTTA, considering CoTTA has already done that. For example, if your source model is not strong, why would it not make sense to update the feature extractor?
> > >
> > > For point W3, what strategies have you explored, and have you done any experiments to justify your choice?
> > >
> > > Also, what is the point of adapting a model every 1 sample on which MDAA really shines? Can you give a practical example to motivate this?

---

### Official Review · Reviewer_3KbA · 2024-11-04

**Soundness:** 2
**Presentation:** 3
**Contribution:** 2
**Rating:** 5
**Confidence:** 4

**Summary:**

This paper proposes a novel multi-modality dynamic analytic adapter (MDAA) approach for addressing the shortcomings of existing multi-modal continual test-time adaptation methods (MM-CTTA). MDAA aims to pay attention to solving the problem of domain shift caused by corruption in cross-domain data, and achieve this goal by integrating analytical classifiers (ACS), dynamic selection mechanism (DSM) and soft pseudo-labeling strategies (SPS). This method has effectively alleviated the superiority of the model by dynamically selecting reliable samples and integrating information of different modals, effectively alleviating problems such as error accumulation, catastrophic forgetting and reliability bias. Finally, the author verified the effectiveness of the proposed method through extensive experiments.

**Strengths:**

1. The paper has a clear motivation for the problem, clearly expresses the existing challenges in the field, and proposes innovative solutions to address them.
2. The method proposed in this article has efficient adaptability, that is, it can adapt without accessing the source data, saving computing resources and protecting data privacy.
3. This method can handle multimodal inputs and has more practical application potential compared to single modal methods.
4. The writing is coherent and logical, serving as a valuable resource for advancing the field.
5. This paper has a certain degree of originality and significance for the development of multimodal fields.

**Weaknesses:**

1. The analysis of the differences between the proposed method and existing methods, as well as the limitations of the analysis method is missing.
2. In the paper, the author proposed three modules (Acs, DSM, and SPS). However, in the ablation experiment, the author lacked experiments on ACs and SPS, as well as DSM and SPS.
3. The running time of the proposed method on the model is also an important part of verifying the superiority of the method. Please supplement the running time of existing methods and all proposed methods in different models to verify whether the method can quickly fine tune and adjust the model.
4. Regarding the content of section 3.3, it still needs to be discussed whether the proposed scenario assumptions and parameter (theta) are consistent with real scenarios.
5. The complexity of the model has not been analyzed, and dynamic selection and soft pseudo labeling strategies may require high computational resources.
6. In the experimental section, only two datasets were used as experimental results to analyze whether the analysis is reliable, and whether there are other similar methods for MM-CTTA that can be compared.

**Questions:**

1. In the model framework of Figure 2, there are image encoders and speech encoders, indicating the existence of neural networks and learnable parameters in the paper. How did the author optimize the entire model, and is there any loss calculation in the model proposed by the author? What is the goal of optimizing the entire model?
2. In the overall model diagram of Figure 2, the information referred to by character B should be represented in the diagram for the convenience of readers' understanding.
3. Unified expression of content. For example, in section 3.4, line 316, subscript XT, t.
4. There is a regularization parameter (gamma) in formulas 6 and 12. The author stated in Appendix C.3 that gamma is within a reasonable range. Please add experiments to verify the rationality of a gamma value of 1.
5. How to fully utilize the complementary and consistent information of multimodal data to deal with corruption data.
6. What are the differences between this method and other methods that address the three core challenges of CTTA problems.
7. How to select pseudo labels and determine their reliability in section 3.4.

---

> ### Author Response · Authors · 2024-11-18
>
> We sincerely appreciate all your attention and comments, here are our replies:
> ***
> **W1:** The analysis of the differences between the proposed method and existing methods, as well as the limitations of the analysis method is missing.
> ***
> Thank you for pointing this out. We have included the differences and the limitations as follows:
>
> While most of existing methods are designed either to solve CTTA or MM-TTA, our method is capable of solving both challenges in CTTA and MM-TTA (i.e. error accumulation, catastrophic forgetting and reliability bias). Our method performs well when evaluated under the highly challenging setting of the superposition of both tasks (i.e., MM-CTTA), where model experiences continuous interleaved modality corruptions.
>
> Specifically, our method is not a simple combination of existing MM-TTA and CTTA methods. To the best of our knowledge, our work is the first to introduce analytic learning to TTA to solve the continual domain shift issue. While most TTA methods update models through backpropagation, our method updates model parameters through a recursive closed-form solution, which **is equivalent to the solution of global optimization on all encountered datasets**, thereby solving the forgetting problem of continual domain shift.
> ***
> **W2:** lacked experiments on ACs and SPS, as well as DSM and SPS.
> ***
> Thank you for your comments on our ablation studies. We have conducted ablation experiments on ACs and SPS to further demonstrate the effectiveness of DSM. As shown in Table 1, incorporating DSM improved accuracy by over 2% compared to configurations without DSM across all experiments.
>
> ACs serve as fundamental solutions for optimizing on both the source and all encountered test domains. Removing ACs would either result in a lack of update rules or violate the CTTA setting, where access to previous data is not permitted. We have theoretically proven that ACs’ solution is equivalent to the solution of joint learning of all encountered data with access to only data in current test stage. Therefore, we are unable to conduct the DSM and SPS experiments.
>
> **Table 1:** Ablation studies on different component combinations.
>
> | Method                    | KS-video | KS-video | VGG-audio | VGG-audio |
> |---------------------------|----------------------------|---------------------------|----------------------------|---------------------------|
> ||$\xrightarrow{}$ |$\xleftarrow{}$ |$\xrightarrow{}$ | $\xleftarrow{}$ |
> | READ                      | 62.32                      | 62.59                     | 23.93                      | 22.39                     |
> | MDAA (ACs)                 | 61.82                      | 62.02                     | 0.47                       | 0.56                      |
> | MDAA (ACs+DSM)             | 63.55                      | **69.54**                 | 34.87                      | 34.85                     |
> | MDAA (ACs+SPS)             | 61.33                      | 67.47                     | 32.50                      | 33.00                     |
> | **MDAA (ACs+DSM+SPS)**     | **65.43**                  | 69.30                     | **35.82**                  | **35.77**                 |
> ***
> **W3:** Need supplement the running time of existing methods and all proposed methods in different models.
> ***
> Thank you for your suggestion. We have conducted additional experiments on runtime and VRAM usage. Since MDAA does not rely on backpropagation, it **requires less computational time and consumes lower memory compared to most existing methods**. While MDAA adapts to test data based on the analytic solution, the computational complexity mainly lies on the matrix inversion, which has a time complexity of $O(N^3)$. Here, N is the feature dimension, which is set as 8000 throughout our experiments. Note that the feature dimension does not increase during CTTA period, thus the complexity of each update keeps consistent.
>
> **Table 2:** Comparison of methods in terms of runtime and memory usage.
> | Method   | KS-video  | KS-video | VGG-audio | VGG-audio |
> |----------|-------------------|----------------------|--------------------|-----------------------|
> |    |time (s) |  memory (MB) |time (s) | memory (MB) |
> | TENT    | 1085              | 37620                | 2571               | 37622                 |
> | SAR      | 2149              | 37620                | 4823               | 39220                 |
> | CoTTA    | 2301              | 55216                | 5283               | 55326                 |
> | EATA     | 889               | 67286                | 2313               | 38300                 |
> | MMTTA   | 2574              | 71732                | 5966               | 68840                 |
> | READ     | 1174              | 9176                 | 2611               | 9178                  |
> | **MDAA** | 1024              | 8784                 | 1682               | 8750                  |

---

> > ### Author Response · Authors · 2024-11-18
> >
> > **W4:** Regarding the content of section 3.3, it still needs to be discussed whether the proposed scenario assumptions and parameter ($\theta$) are consistent with real scenarios.
> > ***
> > The four scenarios described in Section 3.3 encompass all potential cases in real-world situations, representing a comprehensive enumeration of possible label distribution relationships between the leader and follower. For the threshold $\theta$, we determined its value through our ablation experiments in Figure 4(A-B) in main article. Moreover, we also conducted experiments of adjusting its value dynamically during CTTA in Appendix D.3. However, the results from this approach were suboptimal, as shown in Table 3.
> >
> > **Table 3:** Ablation studies on parameter $\theta$.
> >
> > | $\theta$   | KS-video| KS-video | VGG-audio  | VGG-audio  |
> > |------------|---------------------------|--------------------------|----------------------------|---------------------------|
> > |   | $\xrightarrow{}$ | $\xleftarrow{}$ | $\xrightarrow{}$ |  $\xleftarrow{}$ |
> > | Static | 63.55                     | 69.54                    | 34.37                      | 34.35                     |
> > | Dynamic    | 63.49                     | 69.31                    | 34.35                      | 34.30                     |
> > ***
> > **W5:** The complexity of the model has not been analyzed, and dynamic selection and soft pseudo labeling strategies may require high computational resources.
> > ***
> > MDAA consists of four components, each of which we analyze individually:
> >
> > **Forward Inference**: The computational complexity primarily depends on the size of the pre-trained models, which is not influenced by our method.
> >
> > **DSM**: This module is responsible for the prediction selection and has a complexity of $O(N)$, where $N=3$ corresponds to the number of classifiers in MDAA.
> >
> > **SPS**: This module involves sorting label values and has a complexity of $O(N)$, where $N$ refers to the number of classes.
> >
> > **ACs Update**: The complexity of the operation mainly depends on matrix inversion,, which has a time complexity of $O(N^3)$. $N$ is the feature dimension. In our experiments, $N$ is consistently set to 8000.
> >
> > Therefore, the computational complexity of our method is primarily concentrated on the matrix inversion operation, while DSM and SPS require significantly fewer computational resources.

---

> > > ### Author Response · Authors · 2024-11-18
> > >
> > > ***
> > > **W6:** In the experimental section, only two datasets were used as experimental results to analyze whether the analysis is reliable, and whether there are other similar methods for MM-CTTA that can be compared
> > > ***
> > > The dataset used in this study is consistent with those commonly employed in existing video-audio TTA tasks [1–2]. For comparison, we have endeavored to include as many applicable and representative TTA, CTTA, and MM-TTA methods as possible. To the best of our knowledge, only one existing work [3] specifically addresses the MM-CTTA task. However, this work focuses on 3D semantic recognition scenarios and requires data augmentation during training, making it incompatible with the tasks addressed in this paper.
> > >
> > > To make the comparison more comprehensive, **we added an additional methodology** that integrates MM-TTA method READ [1] with model-based stochastic restoration [4] as an additional MM-CTTA approach named **READ+SRM**. Partial comparison results are given in the following tables, while detailed results are given in the revised manuscript.
> > >
> > > **Table 4:** Comparison on **video progressive single-modality corruption** task in terms of classification Top-1 accuracy (\%), with dataset **Kinetics50-C** in severity level 5.
> > >
> > > | Method| Type| Gauss. | Shot.| Impul. | Defoc. | Glass. | Motion. | Zoom. | Snow| Frost |Fog| Bright. |Cont.|Elastic.| Pixel. | Jpeg  | Avg.  |
> > > |----------------|------------|--------|-------|--------|--------|--------|---------|-------|-------|-------|-------|---------|-------|----------|--------|-------|-------|
> > > | READ| MM-TTA|11.92|2.04| 2.03| 2.97| 2.41| 2.46| 2.41| 2.30|2.04|2.04|2.04|2.04|2.04|2.04|2.03|2.86|
> > > | READ+SRM| MM-CTTA|41.69|42.70|42.66| 54.99|54.26|57.95| 56.89| 53.97|52.60 |46.15|61.84|45.30|58.68| 55.52| 54.58 | 51.99 |
> > > |MDAA|MM-CTTA|54.89|55.25|55.32|63.89|62.49|67.26|65.86|64.32|65.31| 61.86|73.20|61.60|67.83 | 69.22 | 68.69| 63.80|
> > >
> > > **Table 5:** Comparison on **video progressive single-modality corruption** task in terms of classification Top-1 accuracy (\%), with dataset **VGGSound-C** in severity level 5.
> > >
> > > | Method| Type| Gauss. | Shot. | Impul. | Defoc. | Glass. | Motion. | Zoom. | Snow  | Frost | Fog   | Bright. | Cont. | Elastic. | Pixel. | Jpeg  | Avg.  |
> > > |----------------|------------|--------|-------|--------|--------|--------|---------|-------|-------|-------|-------|---------|-------|----------|--------|-------|-------|
> > > | READ| MM-TTA| 33.02| 0.33| 0.33| 0.41| 0.33| 0.33 | 0.33| 0.33| 0.33| 0.33| 0.33| 0.33| 0.33|0.33| 0.33| 2.51|
> > > | READ+SRM| MM-CTTA| 41.73| 41.59 | 41.68| 45.02| 45.42| 46.62| 46.09| 44.79| 44.65| 41.93 | 47.13| 39.84 | 46.01    | 44.97  | 45.17| 44.18 |
> > > | MDAA| MM-CTTA| 55.13| 55.29| 55.30 | 56.91| 57.20| 57.78| 57.32| 56.52|56.25| 56.14|58.11| 55.32 |57.06| 56.27|57.39| 56.53|
> > >
> > > [1] Yang M, Li Y, Zhang C, et al. Test-time Adaptation against Multi-modal Reliability Bias[C]//The Twelfth International Conference on Learning Representations. 2024.
> > >
> > > [2] Lei, Jixiang, and Franz Pernkopf. "Two-level test-time adaptation in multimodal learning." ICML 2024 Workshop on Foundation Models in the Wild. 2024.
> > >
> > > [3] Cao H, Xu Y, Yang J, et al. Multi-modal continual test-time adaptation for 3d semantic segmentation[C]//Proceedings of the IEEE/CVF International Conference on Computer Vision. 2023: 18809-18819.
> > >
> > > [4] Wang Q, Fink O, Van Gool L, et al. Continual test-time domain adaptation[C]//Proceedings of the IEEE/CVF Conference on Computer Vision and Pattern Recognition. 2022: 7201-7211.
> > > ***
> > > **Q1:** How did the author optimize the entire model, and is there any loss calculation in the model proposed by the author? What is the goal of optimizing the entire model?
> > > ***
> > > We would like to clarify that all parameters in the encoders remain frozen throughout the entire TTA process. Specifically, we use the pre-trained model CAV-MAE without updating its parameters. **Only ACs’ parameter $\textbf{W} _ {\textup{T},t}$  requires updating**. These parameters are determined by our proposed recursive solution in Theorem 2.
> > >
> > > **Theorem 2:**
> > > $
> > > \hat{\textbf{W}} _ {\textup{T},t} = ({\tilde{\textbf{X}}} _ {\textup{exf},\textup{S}}^\top\tilde{\textbf{X}} _ {\textup{exf}, \textup{S}}\textbf{X} _ {\textup{exf},1:t}^\top \textbf{X} _ {\textup{exf},1:t} \gamma{\textbf{I}})^{-1} ({\tilde{\textbf{X}}}_{\textup{exf}, \textup{S}}^\top{\tilde{\textbf{Y}}} _ {\textup{S}}{\textbf{X}} _ {\textup{exf},1:t}^\top{\bar{\textbf{Y}}} _ {\textup{T},1:t}) =\textbf{P} _ {\textup{T},t}^{-1}\textbf{Q} _ {\textup{T},t}
> > > $
> > >
> > > The loss function used for updates is the **least squares loss** for the regression problem, which we will explicitly highlight in the revised manuscript. Our method adopts a recursive solution to update model parameters, which is **equivalent to the solution of optimizing on all encountered datasets, including data from both source domain and all seen test domains**. This approach effectively addresses the forgetting problem in the continual domain shift setting.

---

> > > > ### Author Response · Authors · 2024-11-18
> > > >
> > > > **Q2:** In the overall model diagram of Figure 2, the information referred to by character B should be represented in the diagram for the convenience of readers' understanding
> > > > ***
> > > > Thank you for your advice, we have added descriptions of character B in both Figure 2 and its caption in the revised manuscript.
> > > > ***
> > > > **Q3:** Unified expression of content. For example, in section 3.4, line 316, subscript XT, t
> > > > ***
> > > > We are very grateful for your careful corrections. The symbols in the manuscript have been rechecked and corrected.
> > > > ***
> > > > **Q4:** There is a regularization parameter (gamma) in formulas 6 and 12. The author stated in Appendix C.3 that gamma is within a reasonable range. Please add experiments to verify the rationality of a gamma value of 1.
> > > > ***
> > > > Thank you for your advice. We have conducted the experiment on gamma as below. **The model performs robustly with a sufficient regularization (i.e., relatively large $\gamma$), while model collapses when the $\gamma$ value is too small**. The experiment is conducted on both Kinetics50 and VGGSound dataset.
> > > >
> > > > **Table 6:** Ablation studies on parameter $\gamma$
> > > > | $\gamma$ | KS-video | KS-video| VGG-audio| VGG-audio|
> > > > |----------|----------------------------|----------------------------|----------------------------|----------------------------|
> > > > ||$\xrightarrow{}$ | $\xleftarrow{}$ | $\xrightarrow{}$ |  $\xleftarrow{}$ |
> > > > | 1e-3     | 2.10| 2.08| 0.51| 0.47|
> > > > | 1e-2     | 2.03| 2.12| 0.31| 0.39 |
> > > > | 1e-1     | 64.87| 69.03| 0.32| 0.37|
> > > > | 1e0      | 65.43| 69.30| 0.36| 0.40 |
> > > > | 1e1      | 65.27| 69.23| 35.82| 35.77|
> > > > | 1e2      | 65.29| 68.95| 35.78| 35.76|
> > > > ***
> > > > **Q5:** How to fully utilize the complementary and consistent information of multimodal data to deal with corruption data.
> > > > ***
> > > > We utilize both the complementary and consistent information through the DSM mechanism.
> > > >
> > > > **Complementarity**: For example, if the video modality of a sample is corrupted, the prediction made by the corresponding video classifier becomes unreliable. Updating the classifier with such prediction would introduce erroneous knowledge. In this case, the DSM selects the more reliable prediction from the audio classifier and uses it as a pseudo label to update the video classifier, enabling it to learn to map corrupted video data to a reliable label, thereby adapting to domain shifts.
> > > >
> > > > **Consistency**: When all classifiers predict the same result, it indicates a high confidence level in the prediction and a lower likelihood of modality corruption. In such case, due to the property of our recursive solution that fully preserves previously learned knowledge, the model does not need to adapt to this test data to ‘consolidate’ what has already been learned. Otherwise, updating the model with such samples could exacerbate class imbalance.

---

> > > > > ### Author Response · Authors · 2024-11-18
> > > > >
> > > > > **Q6:** What are the differences between this method and other methods that address the three core challenges of CTTA problems
> > > > > ***
> > > > > For comparison, we summarize some common methods to address challenges in CTTAs.
> > > > >
> > > > > **Error accumulation:** existing methods mainly adjustment of specific parameters in the model by backpropagation following entropy minimization [2-4] or prediction consistency maximization [1,5-6]. While MDAA migrate this problem by using DSM and SPS. On the one hand, we avoid unnecessary updates to the classifier as much as possible to avoid introducing errors, and on the other hand, we use soft labels to minimize the introduction of errors.
> > > > >
> > > > > **Catastrophic forgetting:** mainstream methods solve this problem by using stochastic restoration mechanism [5] or by generating prototypes from the source dataset [1,6]. Our method updates model parameters by recursively solving the ridge regression problem, such approach is equivalent to the solution of global optimization on all encountered datasets, thereby solving the forgetting problem under CTTA setting without saving any prototype or data.
> > > > >
> > > > > **Reliability bias:** existing works achieve reliable adaptation by modulating the attention module of the ViT block [2,7] based on the property of attention mechanism. Our method is more straightforward by using DSM to select reliable pseudo-labels.
> > > > >
> > > > > [1] Cao H, Xu Y, Yang J, et al. Multi-modal continual test-time adaptation for 3d semantic segmentation[C]//Proceedings of the IEEE/CVF International Conference on Computer Vision. 2023: 18809-18819.
> > > > >
> > > > > [2] Yang M, Li Y, Zhang C, et al. Test-time Adaptation against Multi-modal Reliability Bias[C]//The Twelfth International Conference on Learning Representations. 2024.
> > > > >
> > > > > [3] Wang D, Shelhamer E, Liu S, et al. Tent: Fully Test-Time Adaptation by Entropy Minimization[C]//International Conference on Learning Representations.
> > > > >
> > > > > [4] Niu S, Wu J, Zhang Y, et al. Towards Stable Test-time Adaptation in Dynamic Wild World[C]//The Eleventh International Conference on Learning Representations.
> > > > >
> > > > > [5] Wang Q, Fink O, Van Gool L, et al. Continual test-time domain adaptation[C]//Proceedings of the IEEE/CVF Conference on Computer Vision and Pattern Recognition. 2022: 7201-7211.
> > > > >
> > > > > [6] Xiong B, Yang X, Song Y, et al. Modality-Collaborative Test-Time Adaptation for Action Recognition[C]//Proceedings of the IEEE/CVF Conference on Computer Vision and Pattern Recognition. 2024: 26732-26741.
> > > > >
> > > > > [7] Lei, Jixiang, and Franz Pernkopf. "Two-level test-time adaptation in multimodal learning." ICML 2024 Workshop on Foundation Models in the Wild. 2024.
> > > > >
> > > > > ***
> > > > > **Q7:** How to select pseudo labels and determine their reliability in section 3.4.
> > > > > ***
> > > > > MDAA generates three predictions from three independent ACs for each test sample. The prediction with the highest probability is selected through DSM as the pseudo-label for subsequent updates. This approach is based on the observation that higher prediction probabilities generally indicate that the model has higher confidence in the result, thus more reliable [1-2].
> > > > >
> > > > > [1] Boudiaf M, Mueller R, Ben Ayed I, et al. Parameter-free online test-time adaptation[C]//Proceedings of the IEEE/CVF Conference on Computer Vision and Pattern Recognition. 2022: 8344-8353.
> > > > >
> > > > > [2] Wang Z, Luo Y, Zheng L, et al. In search of lost online test-time adaptation: A survey[J]. International Journal of Computer Vision, 2024: 1-34.
> > > > > ***
> > > > > In light of these clarifications, would you consider increasing your score for our paper? Otherwise, could you let us know any additional changes you would like to see in order for this work to be accepted?

---

> ### Comment · Reviewer_3KbA · 2024-11-19
> **After reading the rebuttals, I am going to low my score to 5.**
>
> Thanks a lot for the rebuttals for my reviews, However, only some of my concerns have been addressed, while there are still some questions that are not answered satisfactorily, thus, I am going to low my score to 5.

---

> > ### Author Response · Authors · 2024-11-19
> >
> > Could you please specify the aspects that are unsatisfactory? We would be willing to provide additional details or clarifications in our response.

---

> > > ### Comment · Reviewer_3KbA · 2024-11-29
> > > **More concerns on the response from the authors**
> > >
> > > Dear authors, the reason why I want to low my score is that some of my concerns are not well addressed in a much clearer way. (1) What I want to see for the response to W1 is that the differences of the machanism or idea of your method to existing methods, instead of the problems or challenges you or others can solve. Additionally, this response lack of the limitations analysis of the proposed method. (2)  In the response to W2, it is good to see you have conducted more ablation experiments , while still missing some analysis on Table 1. (3) In W5, on the complexity analysis of the model, the readers may want to see the complexity comparion of the proposed method with one or two typical similar methods. (4) In Q4, no analysis on the Table 6 is given. I also curious about why the $\gamma$ is not set as 1e3, since there is a setting of $\gamma=1e-3$.

---

> > > > ### Author Response · Authors · 2024-11-29
> > > >
> > > > Thank you for highlighting your concerns. Here are our detailed responses:
> > > > ***
> > > > **(1)** Traditional pseudo-label based TTA methods rely on backpropagation, which optimizes the model on current test data, leading to a locally optimal solution. Therefore, they focus more on how to migrate this limitation by introducing well-designed loss functions or prototypes mechanisms. Our approach, however, updates model parameters using a recursive closed-form solution, **achieving global optimization across all encountered datasets.** This is a major difference compared to previous TTA methods.
> > > >
> > > > Since our method is based on least square error, it is inherently more susceptible to outliers compared to cross-entropy-based approaches. To mitigate this, we introduce the DSM and SPS modules, which leverage intermodal consistency and variability to reduce error accumulation. These modules have proven highly effective, significantly enhancing performance in our experiments.
> > > > ***
> > > > **(2)** The results of (AC+DSM) and (AC+SPS) appear similar because both modules address error accumulation during TTA, proving their effectiveness. However, DSM can leverage intermodal information more effectively, leading to a ~2% performance improvement over SPS across tasks. Moreover, combining DSM and SPS yields even better results.
> > > > ***
> > > > **(3)** We have already provided the **comparison results on time and space complexity in  W3, Table 2**. Could you kindly confirm if this addresses your question? Sorry for any confusion caused.
> > > > ***
> > > > **(4)**
> > > > Based on your suggestion, we have supplemented results with $\gamma$ = 1e3. As shown in Table 6, the model remains robust within the regularization range of 1e1 to 1e2, with performance declining slightly at higher values (when $\gamma$ = 1e3). While the model will also collapses when $\gamma$ is too small.
> > > >
> > > > **Table 6:** Ablation studies on parameter $\gamma$
> > > > | $\gamma$ | KS-video | KS-video| VGG-audio| VGG-audio|
> > > > |----------|----------------------------|----------------------------|----------------------------|----------------------------|
> > > > ||$\xrightarrow{}$ | $\xleftarrow{}$ | $\xrightarrow{}$ |  $\xleftarrow{}$ |
> > > > | 1e-3     | 2.10| 2.08| 0.51| 0.47|
> > > > | 1e-2     | 2.03| 2.12| 0.31| 0.39 |
> > > > | 1e-1     | 64.87| 69.03| 0.32| 0.37|
> > > > | 1e0      | 65.43| 69.30| 0.36| 0.40 |
> > > > | 1e1      | 65.27| 69.23| 35.82| 35.77|
> > > > | 1e2      | 65.29| 68.95| 35.78| 35.76|
> > > > | 1e3      | 59.32| 59.94| 30.36| 30.52|
> > > > ***

---

### Official Review · Reviewer_oE3Z · 2024-11-04

**Soundness:** 2
**Presentation:** 2
**Contribution:** 2
**Rating:** 3
**Confidence:** 4

**Summary:**

The paper proposes a novel method, Multi-modality Dynamic Analytic Adapter (MDAA), for Multi-Modal Continual Test-Time Adaptation (MM-CTTA) tasks. MM-CTTA extends traditional Test-Time Adaptation (TTA) to handle dynamic, multi-modal data with domain shifts caused by corruptions, such as sensor issues or environmental changes. The proposed method main incorporates an additional regression term on the target data for test-time adaptation. Experiments demonstrate that MDAA achieves state-of-the-art performance on MM-CTTA tasks, outperforming existing methods.

**Strengths:**

- Conitnual domain shift poses real challenges to existing deep learning models. Tackling the continual domain shift could improve the robustness and generalization of deep learning models.

- Existing test-time adaptation methods have not addressed the unique challenges in multi-modality corruptions. This paper explored an important but largely dismissed problem.

**Weaknesses:**

- The proposed test-time adaptation method is by simply optimizing the ridge regression problem on both source and target data, as per Eq. (12). The innovation is similar to adding a cross-entropy loss with pseudo labels on target data in regular classification tasks. It is unclear what are the main technical challenges in deploying test-time adaptation methods to multi-modality data.

- Although the paper is motivated by the continual domain shift challenges, the proposed method does not specifically address the challenge of continual domain shift. The overall framework is more like applying self-training to a regression problem.

- The adptation method requires access to source domain data which does not comply with the existing definitions of test-time adaptation or test-time training.

- The dynamic selection mechanism is quite disconnected to the context. I can hardly understand how the dynamic selection mechanism is integrated to the ridge regression problem.

- The manuscript introduced too many subscripts to the notations. It makes the paper hard to follow.

- It may not be necessary to present the closed-form solution to ridge regression with a Threorem. The derivation of closed-form solution is well-known.

**Questions:**

- The authors should provide more discussions on how the proposed method addresses the continual domain shift challenges.

- Further evaluations of the significance of keeping the MSE term on source data is necessary.

---

> ### Author Response · Authors · 2024-11-18
>
> Thanks for your review and valuable comments. Here we address your concerns individually as follows:
> ***
> **W1:** The proposed test-time adaptation method is by simply optimizing the ridge regression problem on both source and target data, as per Eq. (12). The innovation is similar to adding a cross-entropy loss with pseudo labels on target data in regular classification tasks. It is unclear what are the main technical challenges in deploying test-time adaptation methods to multi-modality data
> ***
> We would like to clarify that our method is not simply optimizing on source and target data through standard ridge regression problem. Under the continual test-time adaptation (CTTA) scenario, the model is required to continually adapt to incoming test data stream, which means that the model is forbidden to acquire any previous test data and source data in current test stage. Therefore, **it is unachievable for the model to directly optimize on all encountered source and test data through standard ridge regression.**
>
> Apart from that, our method **extends standard ridge regression to a recursive solution in Theorem 2**. This extension addresses the aforementioned limitation, enabling optimization not only on newly arriving data but also retaining all prior knowledge from both the source and test domain without access to any previous data.
>
> **Theorem 2:**
> $
> \hat{\textbf{W}} _ {\textup{T},t} = ({\tilde{\textbf{X}}} _ {\textup{exf},\textup{S}}^\top\tilde{\textbf{X}} _ {\textup{exf}, \textup{S}}\textbf{X} _ {\textup{exf},1:t}^\top \textbf{X} _ {\textup{exf},1:t} \gamma{\textbf{I}})^{-1} ({\tilde{\textbf{X}}}_{\textup{exf}, \textup{S}}^\top{\tilde{\textbf{Y}}} _ {\textup{S}}{\textbf{X}} _ {\textup{exf},1:t}^\top{\bar{\textbf{Y}}} _ {\textup{T},1:t}) =\textbf{P} _ {\textup{T},t}^{-1}\textbf{Q} _ {\textup{T},t}
> $
>
> The primary challenge of TTA in multimodal data lies in **reliability bias**, which arises from the over-reliance of pre-trained models on specific modalities. For example, as shown in Figure 1(C), when the video modality is corrupted, the performance of both the multimodal and video-only models exhibits a similar downward trend. This occurs because the model disproportionately depends on video information while neglecting audio cues. To address this issue, we propose the Dynamic Selection Mechanism (DSM), which allows the model to dynamically identify and prioritize trustworthy modalities. DSM plays a vital role in selecting reliable samples for each modality-specific classifier, thereby ensuring a robust and effective model update process.
> ***
> **W2&Q1:** How can the method address the challenge of continual domain shift
> ***
> We answer these questions together as they are similar. We address the domain shift problem through our recursive solution in Theorem 1 and DSM module. While existing methods adapt models to current test domain (e.g., data corrupted by Gaussian noise) solely by training the model on these data, this results in overfitting to that specific domain. By contrast, **our method absorbs useful knowledge from the current domain through DSM**, while simultaneously preserves source and encountered domain knowledge during continual adaption via the recursive solution of ACs. Experiment results in Table 1-3 show that performance of our model does not drop during adapting to continual shifting domains compared to other methods. Here we give part of results from Table 1.
>
> **Table 1:** Comparison with SOTA methods on **audio** **progressive single-modality corruption** task in terms of classification Top-1 accuracy (%), using dataset **VGGSound-C** in severity level 5. The best results for each domain are highlighted in **bold**.
>
> | Method | Gauss. | Traff. | Crowd | Rain  | Thund. | Wind  | Avg.  |
> |-----------|--------|--------|-------|-------|--------|-------|-------|
> | Source   | 37.29  | 21.24  | 16.89 | 21.81 | 27.36  | 25.66 | 25.04 |
> | TENT     | 47.88 | 0.68   | 0.28   | 0.28  | 0.28  | 0.28   | 0.28  | 0.35  |
> | SAR       | 50.64 | 16.09  | 4.50   | 4.33  | 3.60  | 12.00  | 5.51  | 7.67  |
> | CoTTA     | 5.33  | 5.85   | 1.35   | 0.52  | 0.53  | 0.57   | 0.38  | 1.53  |
> | EATA       | **40.39** | 31.99 | 31.91 | 32.38 | **39.24** | 33.95 | 34.98 |
> | MMTTA    | 4.50  | 0.41   | 0.33   | 0.33  | 0.33  | 0.33   | 0.33  | 0.34  |
> | READ       | 18.53  | 7.99   | 7.44  | 5.71  | 8.19   | 4.73  | 8.77  |
> | **MDAA** | 38.80  | **34.91** | **34.63** | **34.59** | 37.70  | **35.85** | **36.08** |

---

> > ### Author Response · Authors · 2024-11-18
> >
> > **W3:** The adaptation method requires access to source domain data which does not comply with the existing definitions of test-time adaptation or test-time training.
> > ***
> > We would like to clarify that our method does **NOT** access to any source data during test-time adaption. Our approach only utilizes the source data to initialize the analytic classifiers (ACs) prior to the testing phase. The source data is not used further during the subsequent CTTA process. Such pretraining/initialization on source data is reasonable within the TTA setting and has been widely adopted by several test-time adaptation methods [1-4].
> >
> > [1] Zhang Y, Wang X, Jin K, et al. Adanpc: Exploring non-parametric classifier for test-time adaptation[C]//International Conference on Machine Learning. PMLR, 2023: 41647-41676.
> >
> > [2] Xiong B, Yang X, Song Y, et al. Modality-Collaborative Test-Time Adaptation for Action Recognition[C]//Proceedings of the IEEE/CVF Conference on Computer Vision and Pattern Recognition. 2024: 26732-26741.
> >
> > [3] Cao H, Xu Y, Yang J, et al. Multi-modal continual test-time adaptation for 3d semantic segmentation[C]//Proceedings of the IEEE/CVF International Conference on Computer Vision. 2023: 18809-18819.
> >
> > [4] Yuan L, Xie B, Li S. Robust test-time adaptation in dynamic scenarios[C]//Proceedings of the IEEE/CVF Conference on Computer Vision and Pattern Recognition. 2023: 15922-15932.
> > ***
> > **W4:** The dynamic selection mechanism is quite disconnected to the context:
> > ***
> > We would like to clarify that the Dynamic Selection Mechanism (DSM) is a key component of our method designed for the MM-CTTA setting. DSM addresses two critical challenges in MM-CTTA: (1) the **reliability bias** caused by the unequal reliance of pre-trained models on different modalities, and (2) the **error accumulation** stemming from uncertain pseudo-label predictions. Since analytical classifiers are sensitive to outliers (i.e., corrupted modality), DSM is designed to select more reliable predictions to update each modality-specific classifier.
> >
> > For instance, considering a test sample whose video has been corrupted , the prediction from the video classifier is thus unreliable. Adapting to such video data can degrade the performance of the video classifier. To address this, DSM is designed to prioritize the prediction from the non-corrupted audio modality. It effectively trains the model to map the corrupted data to reliable predicted labels. To experimentally verify its effectiveness, we provide an ablation study on it in Table 2. DSM improves the performance by over 34% in VGG-audio task.
> >
> > **Table 2:** Ablation studies on different component combinations.
> >
> > | Method                    | KS-video | KS-video | VGG-audio | VGG-audio |
> > |---------------------------|----------------------------|---------------------------|----------------------------|---------------------------|
> > ||$\xrightarrow{}$ |$\xleftarrow{}$ |$\xrightarrow{}$ | $\xleftarrow{}$ |
> > | READ                      | 62.32                      | 62.59                     | 23.93                      | 22.39                     |
> > | MDAA (ACs)                 | 61.82                      | 62.02                     | 0.47                       | 0.56                      |
> > | MDAA (ACs+DSM)             | 63.55                      | **69.54**                 | 34.87                      | 34.85                     |
> > | MDAA (ACs+SPS)             | 61.33                      | 67.47                     | 32.50                      | 33.00                     |
> > | **MDAA (ACs+DSM+SPS)**     | **65.43**                  | 69.30                     | **35.82**                  | **35.77**                 |
> > ***
> > **W5:** The manuscript introduced too many subscripts to the notations:
> > ***
> > Thanks for your advice, we will simplify the subscripts to make the manuscript easier to follow.
> > ***
> > **W6:** It may not be necessary to present the closed-form solution to ridge regression with a Theorem:
> > ***
> > Thanks for your advice, Theorem 1 is deriving the P and Q matrices, which retain the source knowledge to support subsequent CTTA tasks. We will change this part from a theorem to a lemma to avoid potential misunderstandings in the manuscript. Regarding Theorem 2, we will emphasize that it extends the standard closed-form solution to a recursive solution that does not require access to past data for clarification.

---

> > > ### Author Response · Authors · 2024-11-18
> > >
> > > **Q2:** Further evaluations of the significance of keeping the MSE term on source data is necessary
> > > ***
> > > Our goal is to obtain the solution to the joint learning of source data and all encountered test data. Keeping the MSE term on source data forms such learning objective. While Eq. (7) and (8) illustrates that the matrix $B_{\textup{S}}$ is used to retain the source knowledge, excluding the learning of source data is equivalent to setting $B_{\textup{S}}$ to zero matrix in the solution. To evaluate the significance of this, we ablated on the $B_{\textup{S}}$ in Table 3 shown as below. **The result shows that without preserving $B_{\textup{S}}$, the performance suffers from a significant drop in all kinds of corruptions**.
> > >
> > > **Eq.(7):**
> > > $
> > > \textbf{P} _ {\textup{S}} = {\tilde{\textbf{X}}} _ {\textup{exf},\textup{S}}^\top\tilde{\textbf{X}} _ {\textup{exf},\textup{S}} + \gamma{\textbf{I}}
> > > $
> > >
> > > **Eq.(8):**
> > > $
> > > \textbf{Q} _ {\textup{S}} = {\tilde{\textbf{X}}} _ {\textup{exf},\textup{S}}^\top{\tilde{\textbf{Y}}} _ {\textup{S}}
> > > $
> > >
> > > **Table 3:** Comparison with SOTA methods on **video** single-modality continual corruption task in terms of classification Top-1 accuracy (%), with dataset **Kinetics50-C** in severity level 5.
> > >
> > > | Method                              | Dir. | Gauss. | Shot. | Impul. | Defoc. | Glass. | Motion. | Zoom. | Snow | Frost | Fog | Bright. | Cont. | Elastic. | Pixel. | Jpeg | Avg. |
> > > |-------------------------------------|-----------|--------|-------|--------|--------|--------|---------|-------|------|-------|-----|---------|-------|----------|-------|------|------|
> > > | MDAA (without $( B _ {\textup{S}} $)) | $\xrightarrow{}$ | 45.13  | 46.71 | 46.21  | 61.69  | 59.44  | 65.80   | 63.56 | 57.74 | 58.87 | 48.88 | 70.66   | 49.21 | 64.93    | 63.59 | 61.81 | 57.62 |
> > > | MDAA                          | $\xrightarrow{}$ | 55.84  | 55.79 | 55.50  | 64.29  | 63.71  | 68.04   | 68.10 | 66.06 | 68.01 | 65.71 | 72.97   | 66.36 | 70.04    | 70.51 | 70.53 | 65.43 |
> > > | MDAA (without $( B _ {\textup{S}} $)) | $\xleftarrow{}$ | 45.37  | 46.6  | 46.42  | 61.73  | 59.58  | 65.91   | 63.56 | 57.74 | 58.83 | 48.95 | 70.84   | 49.28 | 65.14    | 63.7  | 61.84 | 57.70 |
> > > | MDAA                            | $\xleftarrow{}$ | 70.55  | 70.43 | 70.08  | 71.81  | 70.30  | 72.19   | 70.88 | 69.38 | 69.35 | 67.48 | 73.16   | 62.07 | 68.63    | 67.98 | 65.21 | 69.30 |
> > >
> > > ***
> > >
> > > In light of these clarifications, would you consider increasing your score for our paper? Otherwise, could you let us know any additional changes you would like to see in order for this work to be accepted?

---

> ### Author Response · Authors · 2024-11-30
>
> Dear Reviewer,
>
> As the deadline is approaching, we kindly request your feedback or response to our replies. If there are any additional questions or clarifications needed, please let us know, and we are willing to to clarify. Thank you for your time and consideration.
>
> Regards,
>
> The Authors

---

### Author Response · Authors · 2024-11-28
**Kind Request for Reviewer Engagement on Submission 6741**

Dear AC,

With the discussion deadline approaching, we have noticed that only two reviewers have responded to our rebuttal so far. We kindly request your assistance in **prompting reviewers oE3Z and KbiA to engage in the discussion**, as their feedback is crucial for the review process.

Additionally, we would like to bring to your attention that reviewer 3KbA lowered our score without providing specific reasons, despite our submission of four supplementary experiments addressing their initial concerns. We would greatly appreciate it if you could **inquire whether reviewer 3KbA has any remaining questions or concerns**. We are more than willing to provide further clarification or additional experiments if necessary.

Thank you very much for your time and consideration.

Best regards,

Authors

---

### Comment · Area_Chair_5thb · 2024-11-28

Dear Reviewers,

The discussion period has been extended until Dec 2nd. Active participation is highly appreciated and recommended. Thanks for your efforts and contributions to ICLR review process.

Best regards,

Your Area Chair

---

### Meta-Review · Area_Chair_5thb · 2024-12-17

**Metareview:**

The paper proposes a novel method termed Multi-modality Dynamic Analytic Adapter (MDAA) for Multi-Modal Continual Test-Time Adaptation (MM-CTTA) tasks. Reviewers raised concerns about technical differences from existing works, dependency on access to source data (at the beginning of TTA), potential performance drop of using square loss, and lack of analysis for feature space adaptation. At the end of the rebuttal, there were unanswered questions of one reviewer, and the major concerns of the other two reviewers still remained. Thus, this paper can not be accepted by ICLR in its current version.

**Additional Comments On Reviewer Discussion:**

Reviewers raised concerns about technical differences from existing works, dependency on access to source data (at the beginning of TTA), potential performance drop of using square loss, and lack of analysis for feature space adaptation. At the end of the rebuttal, there were unanswered questions of one reviewer, and the major concerns of the other two reviewers remained. Thus, it can not be accepted.

---

### Decision · Program_Chairs · 2025-01-22

Reject